# A dominant negative mitofusin causes mitochondrial perinuclear clusters because of aberrant tethering

Stephanie R Sloat, Suzanne Hoppins ⓘ

In vertebrates, mitochondrial outer membrane fusion is mediated by two mitofusin paralogs, Mfn1 and Mfn2, conserved dynamin superfamily proteins. Here, we characterize a variant of mitofusin reported in patients with CMT2A where a serine is replaced with a proline (Mfn2-S350P and the equivalent in Mfn1, S329P). This serine is in a hinge domain (Hinge 2) that connects the globular GTPase domain to the adjacent extended helical bundle. We find that expression of this variant results in prolific and stable mitochondrial tethering that also blocks mitochondrial fusion by endogenous wild-type mitofusin. The formation of mitochondrial perinuclear clusters by this CMT2A variant requires normal GTPase domain function and formation of a mitofusin complex across two membranes. We propose that conformational dynamics mediated by Hinge 2 and regulated by GTP hydrolysis are disrupted by the substitution of proline at S329/S350 and this prevents progression from tethering to membrane fusion. Thus, our data are consistent with a model for mitofusin-mediated membrane fusion where Hinge 2 supports a power stroke to progress from the tethering complex to membrane fusion.

## Introduction

Mitochondria are dynamic organelles that move, fuse, and divide. Mitochondrial fusion is necessary for maintaining mitochondrial function and cell health (Pernas & Scorrano, 2016; Chan, 2020; Giacomello et al, 2020; Murata et al, 2020). Mitochondria contribute to many cellular functions including energy production, calcium signaling, nutrient sensing, and signaling in innate immunity and cell death (Rambold et al, 2011; Mills et al, 2017; Eisner et al, 2018; Martínez-Reyes & Chandel, 2020; Lim et al, 2021). Mitochondrial structure modulates mitochondrial function. Changes in mitochondrial structure consistent with decreased mitochondrial fusion have been reported in several human diseases, including cardiovascular disease, Alzheimer's disease, diabetes, and cancer (Chen et al, 2011; Zhu et al, 2013; Dai & Jiang, 2019). Recent studies

have shown that restoring balanced mitochondrial dynamics in disease models also corrects mitochondrial morphology and function (Bido et al, 2017; Wang et al, 2017; Rocha et al, 2018; Ferreira et al, 2019; Zhou et al, 2019; Franco et al, 2020).

Mitochondrial fusion and division are mediated by evolutionarily conserved dynamin superfamily proteins (DSPs). The mitochondrial outer membrane fusion machine is composed of two mitofusin paralogs anchored to the outer membrane (Mfn1 and Mfn2) (Santel & Fuller, 2001). Inner membrane fusion is temporally and spatially linked to outer membrane fusion and is mediated by Opa1, also a DSP (Liu et al, 2009; Song et al, 2009). The importance of these processes in organismal health is highlighted by the fact that mutations in *MFN2* or *OPA1* lead to Charcot-Marie-Tooth syndrome Type 2A (CMT2A) and dominant optic atrophy (DOA), respectively (Alexander et al, 2000; Delettre et al, 2000; Züchner et al, 2004; Stuppia et al, 2015). CMT2A is caused by amino acid substitutions throughout Mfn2 and results primarily in a peripheral neuropathy that causes progressive loss of function and sensation in the extremities.

Although Mfn1 and Mfn2 are highly similar and can both individually mediate mitochondrial fusion, they possess important functional distinctions (Santel & Fuller, 2001; Chen et al, 2003; Eura et al, 2003; Ishihara et al, 2004). We have shown that mitochondrial fusion is most efficient if both paralogs are present, highlighting that each mitofusin contributes unique functions to the fusion complex (Hoppins et al, 2011; Engelhart & Hoppins, 2019). Although there are several models of mitofusin topology and structure, the core catalytic features of the mitofusin GTPase domain are comparable with other members of the DSP family (Koshiba et al, 2004; Franco et al, 2016; Qi et al, 2016; Cao et al, 2017; Mattie et al, 2018; Yan et al, 2018; Samanas et al, 2020). Atomic resolution structures of mitofusin minimal catalytic domains (hereafter collectively referred to as mini-mitofusin) revealed that both Mfn1 and Mfn2 possess a globular GTPase domain connected to an extended helical bundle (HB1) by a hinge domain (Hinge 2) (Fig 1A) (Qi et al, 2016; Cao et al, 2017; Yan et al, 2018). A second predicted helical bundle (HB2) and transmembrane region are absent from these structures but are required for membrane fusion as the mini-mitofusins do not mediate lipid mixing. These structural data indicate that Hinge 2 undergoes a large conformational change that moves HB1 relative to the GTPase domain from an open to a closed

University of Washington, Seattle, WA, USA

Correspondence: shoppins@uw.edu

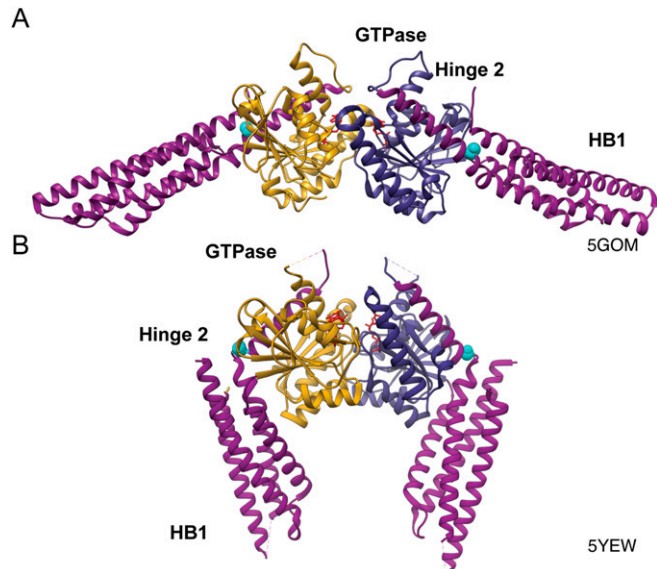

**Figure 1. Structure of Mfn1.**
**(A)** Ribbon structure of Mfn1$_{IM}$ dimer in the open conformation bound to GDP (PDB: 5GOM) with the GTPase domain of one protomer in goldenrod, the second GTPase domain in dark blue, helical bundle 1 (HB1) in magenta, S329 highlighted in cyan, and GDP in red. **(B)** Structure of Mfn1$_{IM}$ dimer in the closed conformation bound to transition-state mimic GDP-BeF$_3^-$ in red (PDB: 5YEW).

state (Fig 1A and B) (Yan et al, 2018). Typical of DSPs, mitofusins form an intermolecular GTPase domain interface (G–G interface) that is required for GTP hydrolysis (Antonny et al, 2016; Qi et al, 2016; Cao et al, 2017; Daumke & Roux, 2017; Yan et al, 2018; Li et al, 2019). Analysis of GTP hydrolysis by the mini-mitofusin constructs revealed that Mfn1 has higher catalytic activity compared with Mfn2 (Li et al, 2019), which likely contributes to their unique membrane fusion properties.

DSPs couple GTP binding and hydrolysis to oligomerization and conformational changes that drive membrane remodeling (Praefcke & McMahon, 2004; Antonny et al, 2016; Ramachandran & Schmid, 2018; Ford & Chappie, 2019; Gao & Hu, 2021). Mitofusin oligomerization occurs in the same membrane, *in cis*, and across two distinct membranes, *in trans*. Both types of assembly are required for efficient mitochondrial fusion, but the molecular details of each are not well defined. Mitofusin oligomerizes in the same membrane upon GTP binding (Ishihara et al, 2004; Anton et al, 2011; Engelhart & Hoppins, 2019; Sloat et al, 2019; Samanas et al, 2020). We have reported that some CMT2A mitofusin variants have impaired GTP-dependent *cis* oligomerization and reduced fusion activity, indicating that *cis* assembly is required for efficient fusion (Engelhart & Hoppins, 2019; Samanas et al, 2020). The mitofusin *trans* complex likely functions as a membrane tether, establishing physical contact between adjacent mitochondrial membranes (Koshiba et al, 2004; Brandt et al, 2016; Engelhart & Hoppins, 2019). Mini-mitofusin inserted into liposomes supported proteoliposome clustering through formation of the G–G interface, leading to the model that mitofusin *trans* complex formation, and thus mitochondrial tethering, requires the G–G interface and GTP hydrolysis (Cao et al, 2017). However, this has not been tested in the context of full-length protein in the mitochondrial membrane. The role of the conformational change mediated by Hinge 2 in membrane fusion is

also not known, but several CMT2A-associated mutations are localized in this region, including one of the most frequently reported variants, Mfn2$^{R94Q}$ (Stuppia et al, 2015). In the context of the mini-mitofusin, this substitution promoted GTP hydrolysis by Mfn2 (Li et al, 2019). The equivalent substitution in Mfn1 modestly reduced dimerization of mini-Mfn1 and impaired the formation of the closed conformational state (Yan et al, 2018; Li et al, 2019). In cells, Mfn2$^{R94Q}$ had no fusion activity alone but can contribute to mitochondrial fusion in the presence of Mfn1$^{WT}$ (Detmer & Chan, 2007). Neuronal expression of Mfn2$^{R94Q}$ resulted in mitochondrial clustering and reduced axonal transport of mitochondria (Misko et al, 2012). A mouse model of CMT2A was generated by overexpression of Mfn2$^{R94Q}$ in neurons, and this recapitulated several neurological features reported in patients, which were reduced with overexpression of Mfn1$^{WT}$ (Zhou et al, 2019). Together, these data highlight that mitofusin Hinge 2 function is important, but the mechanistic basis of its role in membrane fusion is not known.

We had previously performed a functional screen of mitofusin with CMT2A amino acid substitutions at conserved positions in Mfn1 (Engelhart & Hoppins, 2019). We expanded this screen to include additional CMT2A substitutions and observed a striking redistribution of mitochondria to the perinuclear space when either Mfn2$^{S350P}$ or the equivalent Mfn1$^{S329P}$ was expressed in cells. Mfn2$^{S350P}$ changes Hinge 2, and the mutation is inherited as a spontaneous dominant mutation and causes early onset CMT2A and brain lesions (Cho et al, 2007; Chung et al, 2010). Here, we investigate the role of Hinge 2 in mitofusin-dependent mitochondrial fusion by characterizing this disease-associated variant, Mfn2$^{S350P}$, and the equivalent Mfn1$^{S329P}$ variant. We find that these proline variants do not possess fusion activity and block endogenous wild-type mitofusin-mediated mitochondrial fusion. The redistribution of mitochondria to the perinuclear space occurred independently of dynein-based transport but did require that the mitofusin proline variant have a functional GTPase domain. Our data indicate that the mitofusin proline variant forms prolific and stable *trans* complexes throughout the mitochondrial network. Although the variant retains normal GTP hydrolysis, the *trans* complexes do not progress to membrane fusion. We propose that the substitution of proline at Mfn1-S329 or Mfn2-S350 disrupts the conformational dynamics of Hinge 2 within the tethering complex that are induced by GTP hydrolysis and required for lipid mixing. Our data suggest that Hinge 2 supports a power stroke that advances the *trans* complex from tethering to membrane fusion.

# Results

## Expression of Mfn1$^{S329P}$ or Mfn2$^{S350P}$ results in mitochondrial perinuclear clusters

To assess the function of the mitofusin proline variants, we expressed mitofusin with a C-terminal mNeonGreen tag in MEFs lacking either Mfn1 or Mfn2 and visualized mitochondrial morphology by fluorescence microscopy. Mfn1-null and Mfn2-null cells each had fragmented mitochondrial networks when transduced with an empty vector (Fig 2A–D). When wild-type Mfn1 or Mfn2

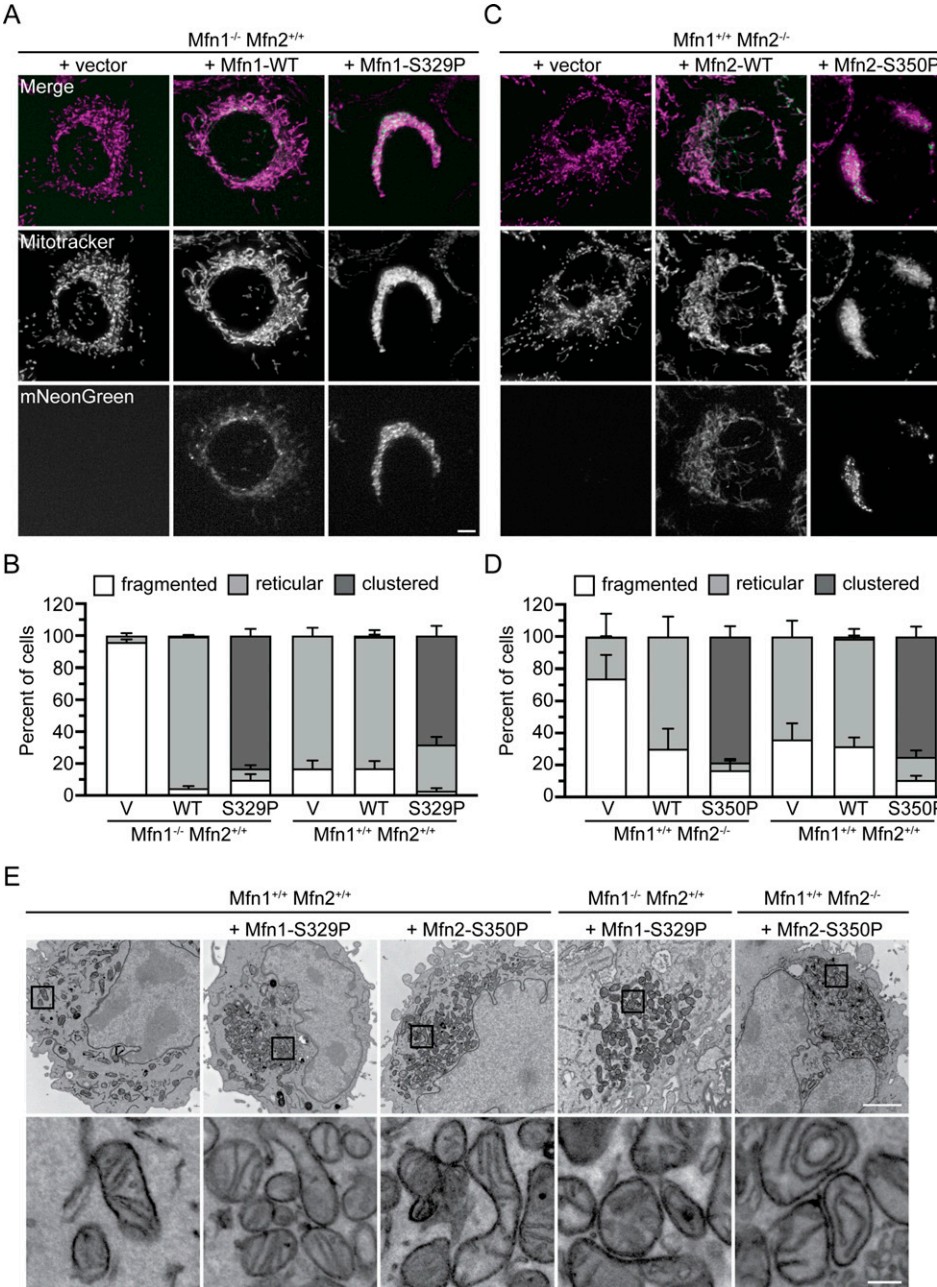

**Figure 2. Expression of Mfn1^S329P or Mfn2^S350P causes perinuclear clustering of mitochondria.**

**(A)** Representative images of Mfn1-null MEFs transduced with an empty vector (V), Mfn1^WT-mNeonGreen, or Mfn1^S329P-mNeonGreen. Mitochondria were labeled with MitoTracker Red CMXRos and visualized with live-cell fluorescence microscopy. Images represent maximum intensity projections. Scale bar = 5 µm. **(A, B)** Quantification of the mitochondrial morphology in the cell lines described in (A) and of wild-type MEFs transduced with an empty vector (V), Mfn1^WT-mNeonGreen or Mfn1^S329P-mNeonGreen. Error bars represent mean ± SEM from n = 3 separate blinded experiments (>100 cells per experiment). **(C)** Representative images of Mfn2-null MEFs transduced with an empty vector (V), Mfn2^WT-mNeonGreen, or Mfn2^S350P-mNeonGreen. Mitochondria were labeled with MitoTracker Red CMXRos and visualized with live-cell fluorescence microscopy. Images represent maximum intensity projections. Scale bar = 5 µm. **(C, D)** Quantification of the mitochondrial morphology in the cell lines described in (C) and wild-type MEFs transduced with an empty vector (V), Mfn2^WT-mNeonGreen, or Mfn2^S350P-mNeonGreen. Error bars represent mean ± SEM from n = 3 separate blinded experiments (>100 cells per experiment). **(E)** Representative electron micrographs of the indicated MEFs cell lines. Scale bar = 2 µm. Lower panels show higher magnification from boxed area in upper panel. Scale bar = 0.2 µm.

(Mfn1^WT or Mfn2^WT) were exogenously expressed in Mfn1-null or Mfn2-null MEFs, respectively, reticular mitochondrial morphology was restored (Fig 2A–D). Interestingly, expression of Mfn1^S329P led to an accumulation of mitochondria in the perinuclear space in 83.1 ± 4.2% of cells (Fig 2A and B). Similarly, Mfn2-null cells expressing Mfn2^S350P had perinuclear mitochondrial clustering in 78.4 ± 6.4% of cells (Fig 2C and D). We observed that Mfn1^WT and Mfn2^WT were evenly distributed along mitochondria, with some mNeonGreen foci, whereas Mfn1^S329P- and Mfn2^S350P-expressing cells had many more mNeonGreen foci (Fig 2A and C). To determine if the mitochondrial redistribution effect was dominant, we expressed the mitofusin proline variants in wild-type MEFs. As expected,

transduction of the empty vector, Mfn1^WT or Mfn2^WT, did not alter the reticular mitochondrial structure, whereas expression of either Mfn1^S329P or Mfn2^S350P resulted in perinuclear clusters of mitochondria (Fig 2B and D).

We also examined the ultrastructure of mitochondria in cells expressing Mfn1^S329P or Mfn2^S350P using transmission electron microscopy. In wild-type MEFs, cristae have a regular electron-dense pattern, and individual mitochondria are spread throughout the cytoplasm (Fig 2E). As expected from our live-cell imaging, in Mfn1-null cells expressing Mfn1^S329P or Mfn2-null cells expressing Mfn2^S350P, mitochondria were found proximal to the nucleus in transmission-electron-microscopy images. The same was true in

wild-type cells expressing either proline variant. The mitochondria in these cells possessed cristae structures that were similar to the wild type, consistent with normal mitochondrial function. Together, these observations of mitochondrial structure and ultrastructure indicate that substitution of Mfn1$^{S329}$ or Mfn2$^{S350}$ with proline results in redistribution of healthy mitochondria to the perinuclear region.

## Mfn1$^{S329P}$ and Mfn2$^{S350P}$ are dominant negative variants that do not support fusion alone and block fusion by wild-type mitofusin

Whether the mitochondria in cells expressing Mfn1$^{S329P}$ or Mfn2$^{S350P}$ are fragmented or reticular within the cluster cannot be discerned by fluorescence microscopy. To distinguish these possibilities, we assessed mitochondrial connectivity by quantifying the diffusion of photoactivatable GFP targeted to the mitochondrial matrix (mt-paGFP). After activation of mt-paGFP inside a small fraction of mitochondria within the network, we followed diffusion and spread of mt-paGFP throughout the network for 50 min. We quantified the fraction of the mitochondrial network with GFP-positive pixels immediately after photoactivation and 50 min later. In cells with low rates of fusion, we expected the fraction to be similar at both time points. In contrast, mitochondrial fusion would promote spread and diffusion of mt-paGFP; thus, the fraction of mitochondria with GFP-positive pixels was expected to increase. We first assessed mitochondrial network connectivity in the Mfn1-null or Mfn2-null background to determine if the mitofusin proline variants supported mitochondrial fusion. In Mfn1-null MEFs transduced with the empty vector, the mitochondrial network was fragmented and the fraction of the network with mt-paGFP–positive pixels increased by 1.30 ± 0.20-fold over 50 min (Fig 3A and C). The results were similar in Mfn2-null cells transduced with the empty vector, where the fraction of the mitochondrial network with mt-paGFP–positive pixels increased 1.51 ± 0.15-fold over the time course (Fig 3B and D). In contrast, Mfn1-null cells expressing Mfn1$^{WT}$ and Mfn2-null cells expressing Mfn2$^{WT}$ both had reticular mitochondria, and after 50 min, the fraction of the network with mt-paGFP increased 2.79 ± .23-fold and 3.04 ± 0.20-fold, respectively (Fig 3A–D). These data are consistent with relatively low fusion rates in either Mfn1-null or Mfn2-null cells and increased fusion activity after expression of wild-type mitofusin. In Mfn1-null cells expressing Mfn1$^{S329P}$, the proportion of the mitochondrial network with mt-paGFP–positive pixels increased by 1.43 ± 0.15-fold over 50 min (Fig 3A and C). Similarly, in Mfn2-null cells expressing Mfn2$^{S350P}$, the fraction of mitochondria with mt-paGFP pixels increased by 1.48 ± 0.08-fold over the time course (Fig 3B and D). Therefore, the diffusion of mt-paGFP in Mfn1-null or Mfn2-null cells expressing either proline variant was comparable to the vector controls. These results indicate that Mfn1$^{S329P}$ and Mfn2$^{S350P}$ are unable to support fusion of mitochondria.

Next, we determined whether mitochondrial fusion by endogenous mitofusin was affected by expression of the proline variants by analyzing diffusion of mt-paGFP in wild-type MEFs expressing either Mfn1$^{S329P}$ or Mfn2$^{S350P}$. Wild-type MEFs transduced with the empty vector have reticular mitochondrial morphology, and the fraction of mt-paGFP pixels in the mitochondrial network increased approximately threefold over 50 min (Figs 3C and D, S1, and S2). Expression of either Mfn1$^{WT}$ or Mfn2$^{WT}$ did not significantly change the diffusion of mt-paGFP in wild-type cells. In contrast, expression

of either Mfn1$^{S329P}$ or Mfn2$^{S350P}$ significantly reduced diffusion of mt-paGFP (1.29 ± 0.09 and 0.97 ± 0.06-fold increase, respectively) (Figs 3C and D, S1, and S2). This low rate of mt-paGFP diffusion was comparable to that observed in Mfn1-null or Mfn2-null cells, which indicates that Mfn1$^{S329P}$ and Mfn2$^{S350P}$ prevent fusion by wild-type mitofusins and therefore possess dominant negative activity.

## Formation of mitochondrial clusters is rapid and specific to mitochondria

Mitochondrial perinuclear clusters have been observed under other conditions, such as in cells drastically overexpressing mitofusin, cells expressing a truncated mitofusin lacking the GTPase domain, cells expressing some CMT2A variants of Mfn2, and cells overexpressing mitoPLD or mitoguardin (Santel & Fuller, 2001; Koshiba et al, 2004; Choi et al, 2006; Huang et al, 2007; Zhang et al, 2016; El Fissi et al, 2018). These mitofusin-dependent perinuclear clusters have been attributed to excessive mitochondrial tethering via mitofusin *trans* complex formation, although this has not been tested. Alternatively, under stress conditions such as hypoxia or heat shock, accumulation of mitochondria in the perinuclear space has been reported to depend on microtubule-based transport by dynein (Al-Mehdi et al, 2012; Agarwal & Ganesh, 2020). Mitochondrial perinuclear clusters induced by mitofusin expression are typically observed 24 h or more after protein expression, whereas acute stress conditions induced mitochondrial clusters within 1–3 h of exposure. To study the molecular basis of the formation of the mitofusin proline variant–induced mitochondrial clusters and to allow temporal control in our experiments, we expressed FLAG-tagged mitofusin from a tetracycline (TET)-inducible promoter in HEK293 Flp-In TREx cells.

In addition to the mitofusin proline variants, we also included the CMT2A variant Mfn2$^{R94Q}$ in these analyses as it was previously reported to result in perinuclear clusters in neurons (El Fissi et al, 2018; Zhou et al, 2019). R94 is also located in Hinge 2, on a different helix than S350, and is therefore a useful reference for general Hinge 2 functions (Fig S3). R94Q is one of the most frequently reported amino acid substitution in CMT2A patients and has been extensively characterized (Detmer & Chan, 2007; Cartoni et al, 2010; Misko et al, 2010, 2012; El Fissi et al, 2018; Zhou et al, 2019; Franco et al, 2020).

We show that cells with vector only or cells expressing Mfn1$^{WT}$, Mfn2$^{WT}$ incubated with 0.2 μg/ml TET for 4 h possessed reticular mitochondria that were indistinguishable from cells incubated without TET (Fig 4A and B). Expression of Mfn1$^{S329P}$, Mfn2$^{S350P}$, or Mfn2$^{R94Q}$ resulted in perinuclear clustering of mitochondria within hours of protein expression, although to slightly different degrees (Figs 4A and B and S4). Specifically, Mfn1$^{S329P}$ expression resulted in perinuclear collapse of the mitochondrial network in almost all cells, whereas the change in distribution was observed closer to half the cells expressing Mfn2$^{S350P}$ or Mfn2$^{R94Q}$. To determine if these distinctions were because of differences in protein expression, we examined protein levels by Western blot analysis of whole-cell protein extract. We found that Mfn1$^{WT}$ and Mfn1$^{S329P}$ were expressed at similar levels, 2.3× and 2×, respectively, compared with endogenous Mfn1 levels observed in vector controls (Fig S4A). Mfn2$^{WT}$, Mfn2$^{S350P}$, and Mfn2$^{R94Q}$ were also expressed at similar levels, at 7.1×, 6.2×, and 6.7× endogenous Mfn2 levels, respectively (Fig S4B). For all mitofusin variants, there were fewer cells with

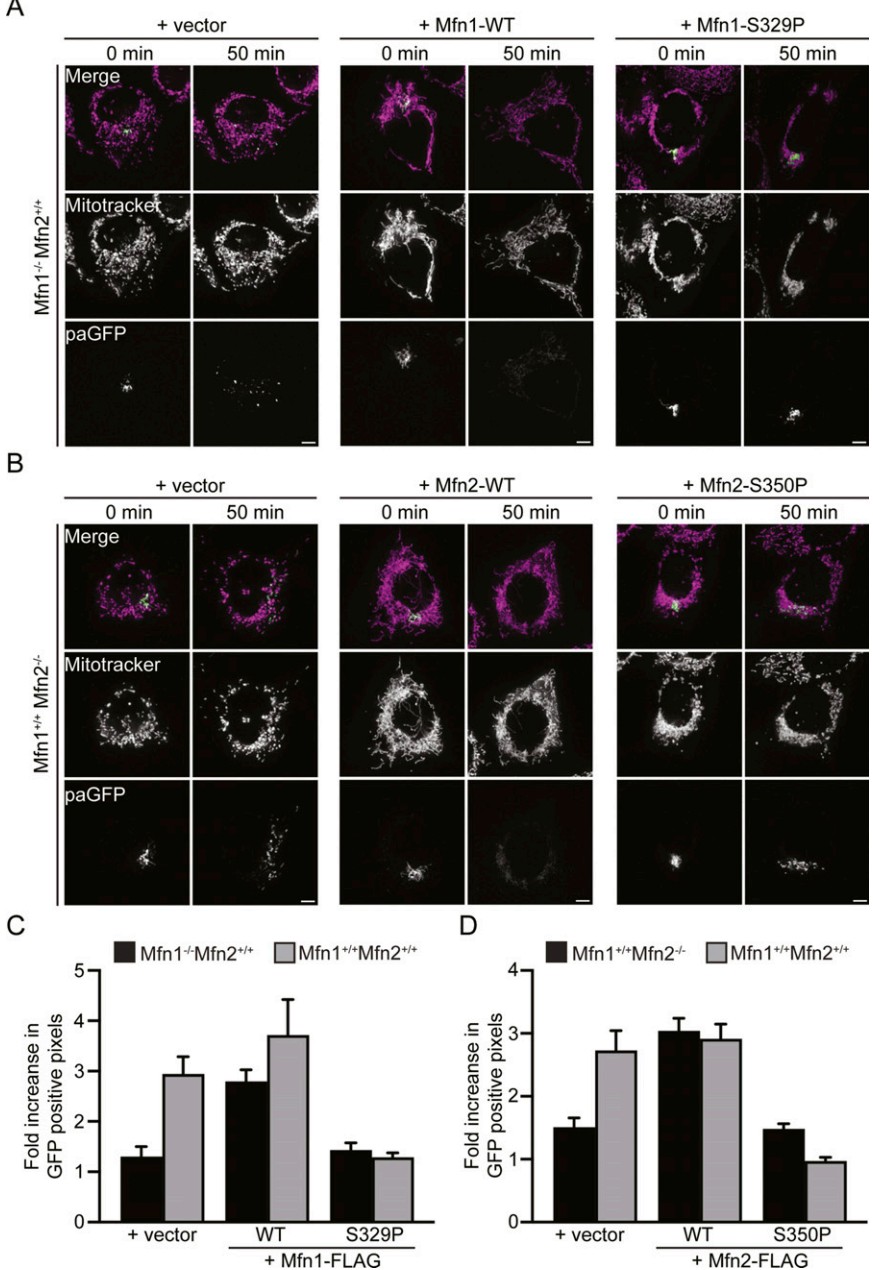

**Figure 3. Mitochondrial clusters induced by proline variants do not have connected mitochondrial network.**

**(A)** Representative images of Mfn1-null MEFs expressing mt-paGFP and transduced with an empty vector (V), Mfn1WT-FLAG, or Mfn1S329P-FLAG either 0 or 50 min after activation of mt-paGFP with a 405-nm laser, which is time = 0. Mitochondria were labeled with MitoTracker Red CMXRos and visualized with live-cell fluorescence microscopy. Images represent maximum intensity projections. Scale bar = 5 $\mu$m. **(B)** Representative images of Mfn2-null MEFs expressing mt-paGFP and transduced with an empty vector (V), Mfn2WT-FLAG, or Mfn2S350P-FLAG either 0 or 50 min after activation of mt-paGFP with a 405-nm laser, which is time = 0. Mitochondria were labeled with MitoTracker Red CMXRos and visualized with live-cell fluorescence microscopy. Images represent maximum intensity projections. Scale bar = 5 $\mu$m. **(A, C)** Quantification of the diffusion of mt-paGFP in the cell lines described in (A) and in wild-type MEFs (Mfn1+/+Mfn2+/+) transduced with an empty vector (V), Mfn1WT-FLAG, or Mfn1S329P-FLAG. Error bars represent mean ± SEM. n = 6/7 cells over two independent experiments. **(B, D)** Quantification of the diffusion of mt-paGFP in the cell lines described in (B) and in wild-type MEFs (Mfn1+/+Mfn2+/+) transduced with an empty vector (V), Mfn2WT-FLAG, or Mfn2S350P-FLAG. Error bars represent mean ± SEM. N = 9–11 cells over three independent experiments.

mitochondrial clusters observed at 2 h and more observed at 6 h, although the protein expression did not change significantly at these time points (Fig S4C and D). Together, our results demonstrate that the redistribution of mitochondria to the perinuclear space occurs relatively rapidly after expression of the mitofusin proline variant and that Mfn1S329P is more effective at evoking the redistribution of mitochondria.

We visualized lysosomes and ER in cells expressing the mitofusin proline variants after induction by TET to determine if the expression of Mfn1S329P or Mfn2S350P alters the distribution of other organelles. In cells expressing Mfn1WT or Mfn2WT, lysosomes appear as small puncta distributed throughout the cytoplasm. This distribution was not changed in cells expressing Mfn1S329P or Mfn2S350P

(Fig S5A). TREx cells were transfected with Sec61-GFP to visualize the structure of the ER, which was similar in cells expressing wild-type mitofusin and the mitofusin proline variant (Fig S5B). These data demonstrate that the mitofusin proline variants do not induce global perinuclear clustering of organelles but have a specific effect on mitochondrial distribution.

### Perinuclear clustering of mitochondria is not dependent on mitochondrial transport by dynein

Our data could be consistent with a model in which mitochondria that possess mitofusin proline variants have altered microtubule-

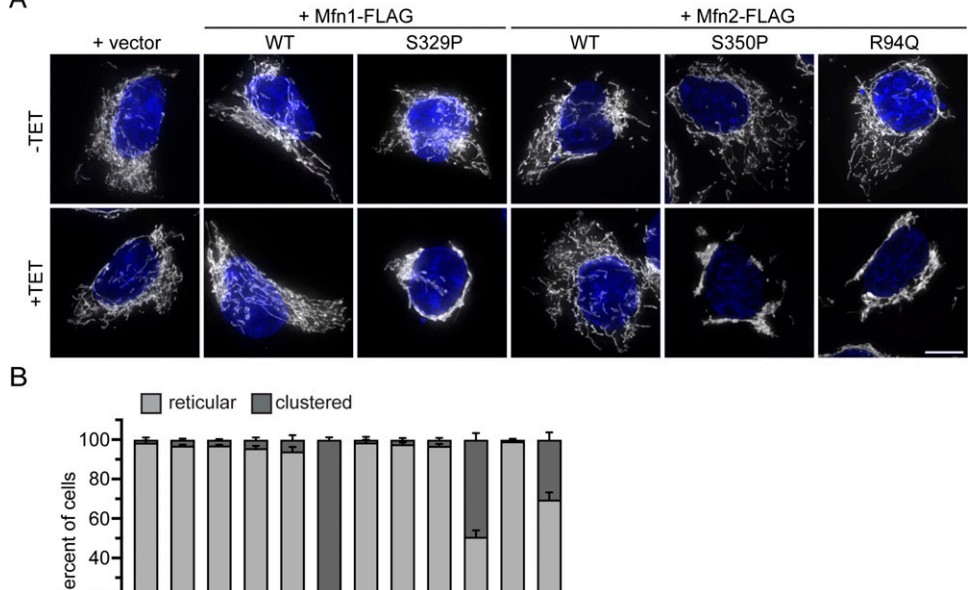

**Figure 4. Mitofusin proline variants induce rapid perinuclear clustering of mitochondria.**

**(A)** Representative images of Flp-In TREx HEK293 expressing the indicated mitofusin variant after incubation with 0.2 µg/ml TET for 4 h. Mitochondria were labeled with MitoTracker Red CMXRos, and nuclei were labeled with NucBlue and visualized with live-cell fluorescence microscopy. Images represent maximum intensity projections. Scale bar = 5 µm. **(A, B)** Quantification of the mitochondrial morphology in the cell lines described in (A). Error bars represent mean ± SEM from n = 3 separate blinded experiments (>100 cells per experiment).

based transport, driving mitochondria toward the perinuclear space. Adapter proteins on the mitochondrial surface recruit dynein, which directs retrograde movement of mitochondria on microtubules toward the nucleus, and kinesin, which is responsible for anterograde movement toward the cell periphery (Kruppa & Buss, 2021). To determine if mitochondrial perinuclear clusters require dynein-dependent transport on microtubules, we induced expression of mitofusin in the TRex cells in the presence of nocodazole, which depolymerizes microtubules. Immunohistochemical staining confirmed that microtubules were fully depolymerized (Fig S6A). Cells expressing Mfn1$^{WT}$ or Mfn2$^{WT}$ had predominantly reticular mitochondria that were distributed throughout the cytosol in the presence or absence of nocodazole (Fig 5A and B). In contrast, in cells expressing Mfn1$^{S329P}$, Mfn2$^{S350P}$, or Mfn2$^{R94Q}$, we observed formation of perinuclear clusters comparable to control cells without nocodazole (Fig 5A and B). Interestingly, the shape of the mitochondrial cluster is slightly different in nocodazole-treated cells where the mitochondria appear to be more compactly arranged, and for Mfn2 variants, slightly more cells had mitochondria in perinuclear clusters. We have previously shown that expression of Mfn1$^{F202L}$ in Mfn1-null cells also resulted in accumulation of mitochondria in the perinuclear region (Engelhart & Hoppins, 2019). In this case, treatment of cells with nocodazole was sufficient to redistribute the mitochondria throughout the cytoplasm, similar to cells overexpressing Drp1 (Smirnova et al, 1998). This was not true for cells expressing Mfn1$^{S329P}$, Mfn2$^{S350P}$, or Mfn2$^{R94Q}$, which suggests that these inter-mitochondrial contacts are stable.

These data suggest that microtubule transport is not required for redistribution of mitochondria to the perinuclear space upon expression of these mitofusin variants. To further support this

conclusion, we knocked down expression of dynein heavy chain (DHC) using shRNA in the TET-inducible cells. To determine shRNA efficiency, we performed Western Blot analysis of whole-cell lysates and observed that DHC protein levels were reduced 80–90% compared with cells treated with control shRNA against LacZ (Fig S6B). Mitochondrial morphology in cells expressing Mfn1$^{WT}$ or Mfn2$^{WT}$ looked similar in DHC knockdown cells and the LacZ controls. Expression of Mfn1$^{S329P}$, Mfn2$^{S350P}$, or Mfn2$^{R94Q}$ led to perinuclear clustering of mitochondria in cells with reduced DHC expression and LacZ controls (Fig 5C and D). We note that there is a slight trend toward fewer clusters with reduced expression of DHC, suggesting that dynein activity may contribute to perinuclear localization of mitochondria to some extent, but it is not statistically significant. Together, our data indicate that mitochondrial perinuclear clustering because of CMT2A mitofusin variants is not dependent on dynein-directed microtubule-based transport.

## Mfn1$^{S329P}$- and Mfn2$^{S350P}$-induced mitochondrial clusters requires a functional GTPase domain

Our data demonstrate that these mitofusin proline variants do not support fusion and block fusion by wild-type mitofusin while also leading to the accumulation of mitochondria in the perinuclear space independent of microtubule-based transport. This is consistent with a model where the proline variant engages in abundant *trans* complexes that are stable and cannot progress to membrane fusion. We evaluated the role of GTP binding and hydrolysis in the function of the mitofusin proline variants to test this model and determine what catalytic functions are required to evoke mitochondrial perinuclear clusters.

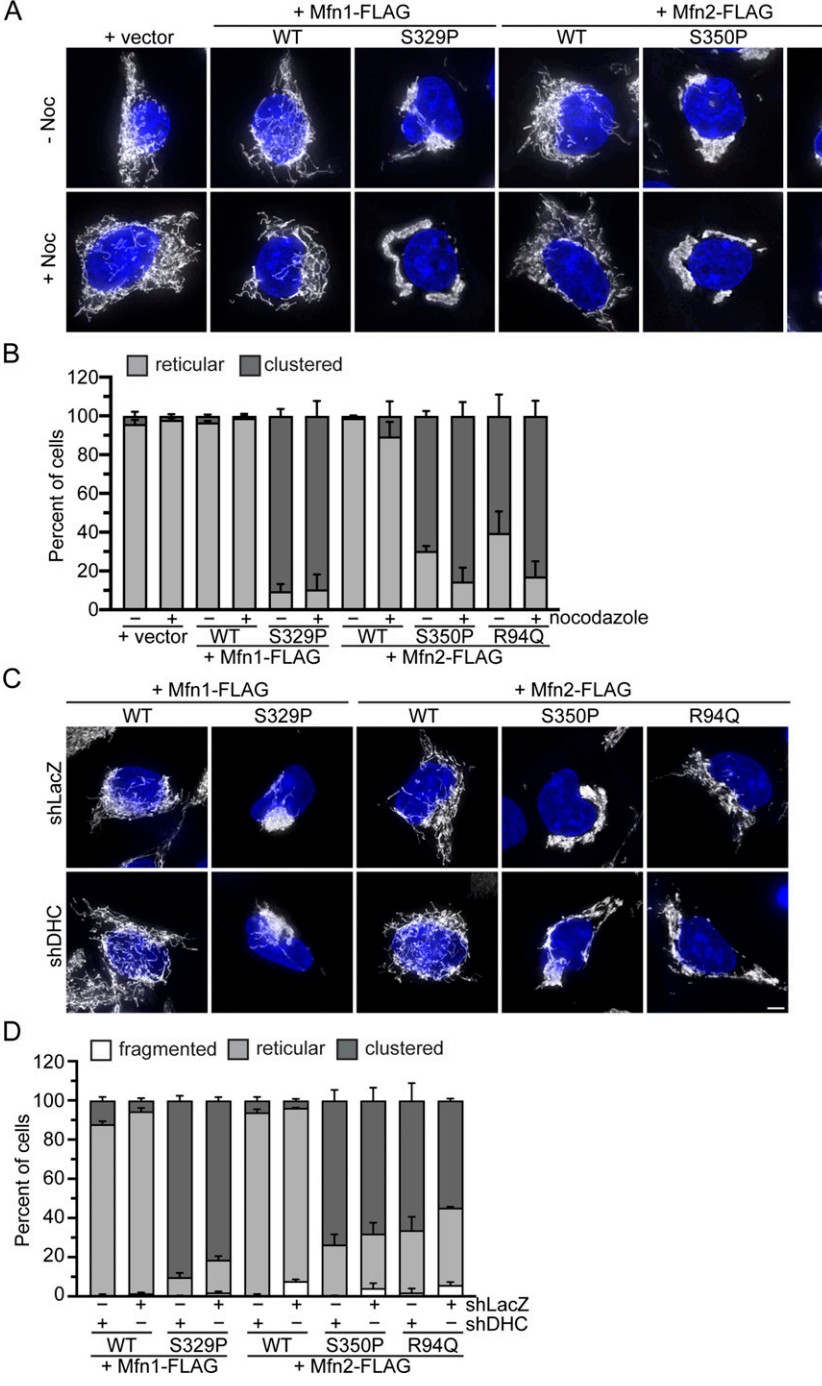

**Figure 5. Perinuclear clustering of mitochondria by the proline variant is not dependent on dynein-mediated microtubule–based transport.**
**(A)** Representative images of Flp-In TREx HEK293 expression indicated the mitofusin variant after incubation with 0.2 μg/ml TET for 4 h in the presence or absence of 5 nM nocodazole. Mitochondria were labeled with MitoTracker Red CMXRos, and nuclei were labeled with NucBlue and visualized by live-cell fluorescence microscopy. Images represent maximum intensity projections. Scale bar = 5 μm. **(A, B)** Quantification of the mitochondrial morphology in the cell lines described in (A). Error bars represent mean ± SEM from n = 3 separate and blinded experiments (>100 cells per experiment). **(C)** Representative images of Flp-In TREx HEK293 with shLacZ or shDHC and expressing the indicated mitofusin variant after incubation with 0.2 μg/ml TET for 4 h. Mitochondria were labeled with MitoTracker Red CMXRos, and nuclei were labeled with NucBlue and visualized by live-cell fluorescence microscopy. Images represent maximum intensity projections. Scale bar = 5 μm. **(C, D)** Quantification of the mitochondrial morphology in the cell lines described in (C). Error bars represent mean ± SEM from n = 3 separate and blinded experiments (>100 cells per experiment).

To measure the enzymatic activity of Mfn1$^{S329P}$, we used the mini-mitofusin Mfn1$_{IM}$ that consists of residues 1–365 and 696–741 connected by a flexible linker (Cao et al, 2017). We found the kinetics of GTP hydrolysis was similar between Mfn1$_{IM}^{WT}$ and Mfn1$_{IM}^{S329P}$ (Fig S7A and B). This indicates that the proline substitution does not affect GTP binding or hydrolysis. Given that GTP hydrolysis is triggered by formation of the intermolecular GTPase domain interface, our data also indicate that the formation of the G–G interface is not altered by the proline substitution.

To determine the role of the GTPase domain in mitochondrial clustering induced by these mitofusin proline variants in cells, we altered GTPase activity in either Mfn1$^{S329P}$ or Mfn2$^{S350P}$ with amino acid substitutions shown to disrupt distinct steps in the catalytic cycle (Fig S7C). These experiments were guided by functional information uncovered from atomic resolution structures of the mini-mitofusin constructs, which have provided significant mechanistic insight into GTP binding and hydrolysis. Of note, there are paralog-specific differences and variations between

individual Mfn1 structures that raise questions about the specific role(s) of some features of the GTPase domain. Therefore, we used several amino acid substitutions in the GTPase domain to disrupt the catalytic cycle at different stages. We expressed these as full-length variants in Mfn1-null or Mfn2-null cells so that we could assess mitochondrial fusion activity simultaneously.

The P-loop of DSPs has a key lysine residue that has been well documented to allow GTP binding but disrupts GTP hydrolysis when substituted with alanine (van der Bliek et al, 1993; Smirnova et al, 1998; Chappie et al, 2010; Tornabene et al, 2020). When this substitution is made in Mfn1 (Mfn1$^{K88A}$), the variant lacks GTPase activity, and another substitution at this site (Mfn1$^{K88T}$) did not

support fusion in Mfn1-null cells (Chen et al, 2003; Qi et al, 2016). As expected, we found that expression of Mfn1$^{K88A}$ in Mfn1-null cells failed to rescue mitochondrial morphology as 97.9 ± 1.3% of cells possessed fragmented mitochondria (Fig 6A and B). When Mfn1$^{K88A-S329P}$ was expressed in Mfn1-null cells, the mitochondrial network also remained fragmented in most cells (75.1 ± 3.0%). Significantly, very few of the cells possessed clustered mitochondria (22.9 ± 2.5%) compared with cells expressing Mfn1$^{S329P}$ (75.9 ± 3.4%) (Fig 6A and B). Similarly, most Mfn2-null cells expressing Mfn2$^{K109A}$ had fragmented mitochondrial networks (Fig 6C and D). In contrast, more than half of the Mfn2-null cells expressing Mfn2$^{K109A-S350P}$ had reticular mitochondria (62.7 ± 12.3%), and very few exhibited

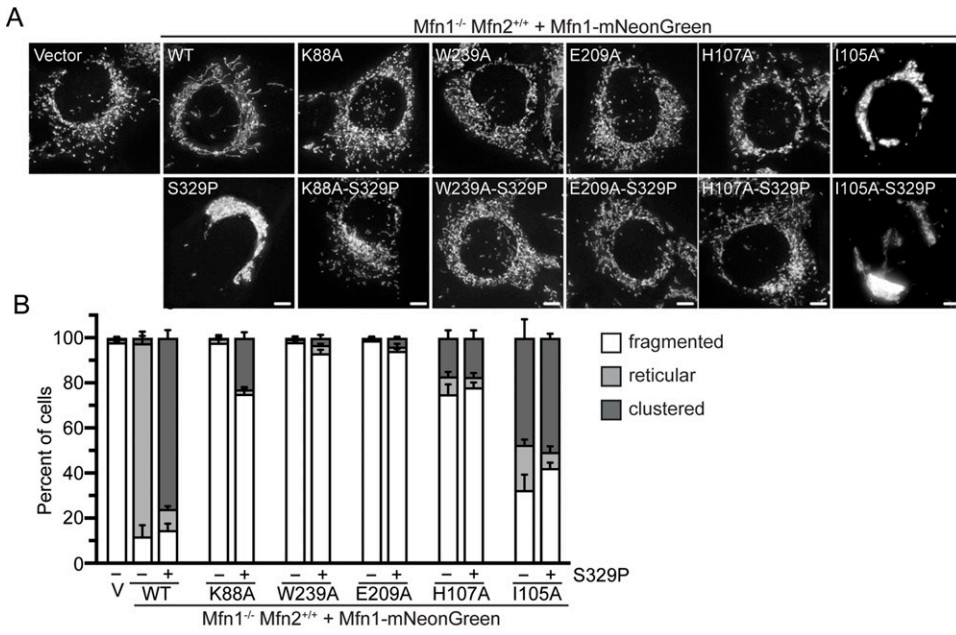

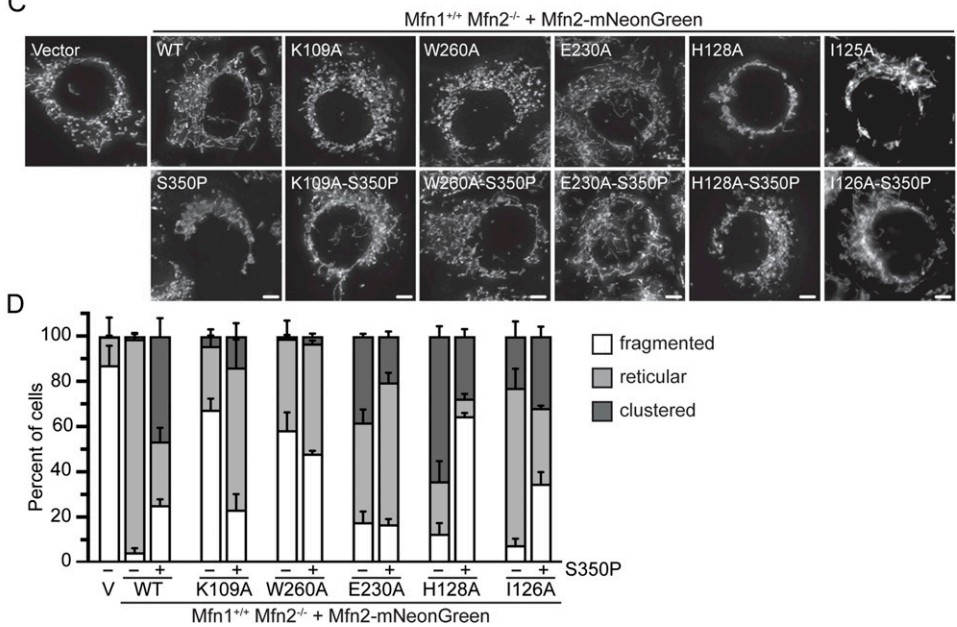

**Figure 6. Mitofusin proline variants require GTPase activity to induce perinuclear clustering of mitochondria.**
**(A)** Representative images of Mfn1-null MEFs expressing the indicated Mfn1-mNeonGreen variant. Mitochondria were labeled with MitoTracker Red CMXRos and visualized with live-cell fluorescence microscopy. Images represent maximum intensity projections. Scale bar = 5 µm. **(A, B)** Quantification of the mitochondrial morphology in the cell lines described in (A). Error bars represent mean ± SEM from n = 3 separate blinded experiments (>100 cells per experiment). **(C)** Representative images of Mfn2-null MEFs expression indicated the Mfn2-mNeonGreen variant. Mitochondria were labeled with MitoTracker Red CMXRos and visualized with live-cell fluorescence microscopy. Images represent maximum intensity projections. Scale bar = 5 µm. **(C, D)** Quantification of the mitochondrial morphology in the cell lines described in (C). Error bars represent mean ± SEM from at least n = 3 separate blinded experiments (>100 cells per experiment).

perinuclear clustering of mitochondria (Fig 6C and D). The fact that we observe Mfn2-null cells with reticular mitochondrial network upon expression of Mfn2$^{K109A-S350P}$ indicates that this variant can support fusion in the presence of Mfn1$^{WT}$ and reveals that the proline substitution at S350 does not block fusion activity of Mfn2 in this context. These data show that GTP hydrolysis is necessary for perinuclear clustering of mitochondria in cells expressing Mfn1$^{S329P}$ or Mfn2$^{S350P}$.

To determine if GTP binding is necessary for clustering of mitochondria by the proline variants, we made Mfn1$^{W239A}$, which has been shown to abolish GTP binding in mini-Mfn1 (Cao et al, 2017). A subsequent atomic structure of mini-Mfn1 suggested that W239 may also have a role in stabilizing the G–G interface (Yan et al, 2018). In the mini-Mfn2 structure, the equivalent tryptophan W260 was facing away from the nucleotide binding pocket, which the authors proposed facilitates GTP binding (Li et al, 2019). In Mfn1-null cells expressing Mfn1$^{W239A}$, mitochondria were fragmented in most of the cells, similar to previous reports (Cao et al, 2017). Mfn1-null cells expressing Mfn1$^{W239A-S329P}$ also possessed primarily fragmented mitochondrial networks (Fig 6A and B). Expression of Mfn2$^{W260A}$ or Mfn2$^{W260A-S350P}$ in Mfn2-null cells resulted in a partial rescue of mitochondrial morphology, consistent with previous reports that the W260A variant can support some mitochondrial fusion (Engelhart & Hoppins, 2019; Sloat et al, 2019). Neither Mfn2$^{W260A}$ nor Mfn2$^{W260A-S350P}$ induced perinuclear clustering in Mfn2-null cells (Fig 6C and D). These data indicate that GTP binding is also necessary for perinuclear clustering of mitochondria induced by expression of the proline variants.

The intermolecular GTPase domain interface is stabilized by an intermolecular salt bridge between E209 and R238 in Mfn1 (E230 and R259 in Mfn2). Disruption of this salt bridge by alanine substitutions allows GTP binding but inhibits GTP hydrolysis, resulting in loss of mitochondrial fusion activity as evidenced by fragmented mitochondria when the mitofusin variants were expressed in Mfn1/2-null cells (Cao et al, 2017). To determine if the G–G interface is necessary for mitochondrial clustering induced by the mitofusin proline variants, we tested the effect of expression of Mfn1$^{E209A-S329P}$ on mitochondrial morphology in Mfn1-null cells. We observed that most cells expressing either Mfn1$^{E209A}$ or Mfn1$^{E209A-S329P}$ possessed fragmented mitochondria, consistent with no mitochondrial fusion activity (Fig 6A and B). In cells expressing either Mfn2$^{E230A}$ or Mfn2$^{E230A-S350P}$, slightly more than half of the cells had reticular mitochondria, indicative of mitochondrial fusion activity for both variants when Mfn1 is also present. Mfn2$^{E230A}$ expression also resulted in some cells with perinuclear clusters of mitochondria, although less than Mfn2$^{S350P}$ alone. Interestingly, Mfn2-null cells expressing Mfn2$^{E230A-S350P}$ possessed fewer mitochondrial perinuclear clusters compared with cells expressing Mfn2$^{E230A}$ or Mfn2$^{S350P}$ (Fig 6C and D). Therefore, formation of the G–G interface is also necessary for perinuclear clustering of mitochondria caused by Mfn1$^{S329P}$ or Mfn2$^{S350P}$.

Mfn1-H107 has been suggested to have a role in charge compensation within the GTP binding pocket, and substitution of alanine at H107 permitted GTP to bind but prevented hydrolysis (Cao et al, 2017). Of note, other structural studies suggest that H107 points away from the binding pocket, which could imply a different or additional role for H107 in catalysis, such as formation of the G–G

interface (Yan et al, 2018). To determine if Mfn1-H107 stabilizes the G–G interface, we purified Mfn1-IM$^{H107A}$ and tested for its ability to dimerize in the presence of the transition-state mimic GDP·BeF$_3^-$ by sucrose gradient centrifugation. Although Mfn1-IM$^{WT}$ was a monomer in the presence of GDP alone, incubation with GDP·BeF$_3^-$ shifted the protein size to that consistent with a dimer (Fig S8). In contrast, Mfn1-IM$^{H107}$ did not dimerize under the same conditions (Fig S8). Therefore, this variant has a weakened dimer interface, which likely impairs GTP hydrolysis. It has been reported that expression of either Mfn1$^{H107A}$ or Mfn2$^{H128A}$ in cells lacking both mitofusins (Mfn1/2-null cells) resulted in an abnormal mitochondrial network, which included some perinuclear clusters (Cao et al, 2017). When we expressed Mfn1$^{H107A}$ in Mfn1-null cells, we observed primarily fragmented mitochondria (74.9 ± 4.4%) with a small degree of mitochondrial clustering in the perinuclear region (17.1 ± 3.3%). The difference between our data and previous results could be because of expression in Mfn1-null cells rather than Mfn1/2-null cells or a difference in the degree of overexpression. With expression of Mfn1$^{H107A-S329P}$ in Mfn1-null cells, we observed that 78.1 ± 2.2% of cells had fragmented mitochondria and 17.3 ± 3.3% had mitochondrial clustering, comparable to Mfn1$^{H107A}$ alone (Fig 6A and B). With expression of Mfn2$^{H128A}$ in Mfn2-null cells, we observed perinuclear clustering in more than half of the cells (64.1 ± 4.4%), similar to reports after expression in Mfn1/2-null cells (Cao et al, 2017). Interestingly, few Mfn2-null cells expressing Mfn2$^{H128A-S350P}$ possessed clustered mitochondria (27.6 ± 3.0%) (Fig 6C and D). These data are also consistent with the conclusion that destabilization of the G–G interface significantly reduces mitochondrial perinuclear clusters induced by Mfn1$^{S329P}$ or Mfn2$^{S350P}$.

Mfn1-I105 is in switch I and was found to pack against W329 in the pairing molecule of a mini-Mfn1 dimer (Yan et al, 2018). Yan et al (2018) assessed the role of this residue in catalytic activity by substituting I105 with alanine, and although dimerization was only modestly impaired, GTPase activity was greatly diminished (Yan et al, 2018). Therefore, this substitution was used to assess the role of GTP hydrolysis in the formation of perinuclear clusters by the mitofusin proline variants. We first expressed Mfn1$^{I105A}$ in Mfn1-null cells and observed cells with either mitochondrial perinuclear clusters or fragmented mitochondrial networks (47.8 ± 8.2% and 32.5 ± 6.8%, respectively). Although mitochondrial perinuclear clusters were observed less frequently in cells expressing Mfn1$^{I105A}$ compared with Mfn1$^{S329P}$, we wanted to determine if Mfn1$^{I105A}$ was mediating mitochondrial fusion within the mitochondrial clusters. Therefore, we monitored the diffusion of mt-paGFP in the mitochondrial clusters observed in Mfn1-null cells expressing Mfn1$^{I105A}$. After 50 min, the increase in the fraction of the mitochondrial network that possessed GFP-positive pixels was comparable between vector controls and cells expressing Mfn1$^{I105A}$ (1.3 ± 0.09-fold and 1.5 ± 0.05-fold, respectively) (Fig S9). This indicates that Mfn1$^{I105A}$ does not mediate mitochondrial fusion and that the mitochondrial clusters are composed of discontinuous organelles. We then asked if Mfn1$^{I105A}$ possessed dominant negative activity by monitoring the diffusion of mt-paGFP in wild-type cells expressing this variant. In wild-type MEFs transduced with the empty vector, the fraction of GFP-positive pixels increased about twofold after 50 min (2.0 ± 0.12-fold) (Fig S9). In contrast, in the mitochondrial perinuclear clusters observed in wild-type cells expressing Mfn1$^{I105A}$, there was little

increase in the fraction of the network with GFP-positive pixels (1.3 ± 0.1-fold) (Fig S9). This indicates that, in the context of the mito-chondrial clusters, Mfn1[I105A] blocks fusion activity by endogenous wild-type mitofusins. Together, this analysis of Mfn1[I105A] suggests that formation of the Mfn1 G–G interface is sufficient to promote some mitochondrial clustering and that GTP hydrolysis is not re-quired. Expression of Mfn1[I105A-S329P] in Mfn1-null cells resulted in either perinuclear mitochondrial clusters or mitochondria that remained fragmented, comparable to Mfn1[I105A] alone. This is consistent with a model where mitochondrial tethers by Mfn1[I105A] or Mfn1[S329P] are initiated by the formation of the G–G interface and that subsequent GTP hydrolysis by Mfn1[S329P] results in a more severe phenotype. For Mfn2, we found that most Mfn2-null cells that were expressing Mfn2[I126A] possessed a reticular mitochondrial network (70 ± 8.6%) (Fig 6C and D). In Mfn2-null cells expressing Mfn2[I126A-S350P], we observed a similar fraction of cells with either fragmented, reticular, or clustered mitochondrial networks (Fig 6C and D). These data indicate that disruption of GTP hydrolysis sig-nificantly decreases the mitochondrial clustering caused by ex-pression of Mfn2[S350P]. In sum, these data are consistent with a model where GTP binding and subsequent formation of the in-termolecular GTPase domain interface are both required for mitofusin tethering and that GTPase activity in the proline variants is required for their prolific mitochondrial clustering phenotype.

### Nucleotide-dependent cis-assembly of Mfn1[S329P] is reduced

Mitofusins assemble into at least two distinct oligomers upon GTP binding. We have previously shown that other CMT2A-associated mitofusin variants with defects in mitochondrial fusion have im-paired GTP-dependent assembly (Engelhart & Hoppins, 2019; Samanas et al, 2020). To assess the GTP-dependent assembly of Mfn1[S329P], we performed blue native–PAGE of solubilized mito-chondria that possess either Mfn1[WT]-FLAG or Mfn1[S329P]-FLAG. In the absence of additional nucleotide, Mfn1[WT] exists predominantly as a single species that migrates with an estimated molecular weight of 200 kD, which is consistent with a *cis* dimer (arrowhead, Fig 7A and B). We incubated mitochondria in the presence of the non-hydrolyzable GTP analog, GMP-PNP, which shifts mitofusin from the dimer into two larger species that migrate at ~320 and ~450 kD (Fig 7A and B, open and closed arrows). In the absence of nucle-otide, Mfn1[S329P] is indistinguishable from the wild type, indicating that the *cis* dimer is not altered. In contrast, in the presence of GMP-PNP, very little of Mfn1[S329P] was found in either the ~320- or ~450-kD species (Fig 7A and B, open and closed arrows, respectively). Given that in the context of Mfn1[IM], the proline variant can bind and hydrolyze GTP, this indicates that Mfn1[S329P] is deficient for GTP-dependent *cis*-oligomerization.

### Mfn1[S329P]- and Mfn2[S350P]-induced mitochondrial perinuclear clusters require mitofusin *trans* assembly

Our data are consistent with a model where these mitofusin var-iants cause unrestricted mitochondrial tethering in the cell such that virtually all mitochondria are connected via mitofusin com-plexes in *trans*. We predicted that if stable *trans* interactions lead to mitochondrial clusters, we should block their formation by

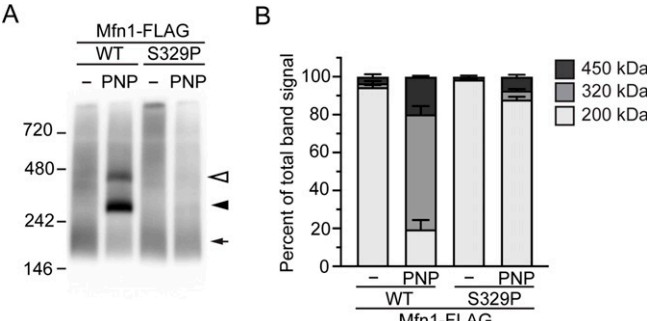

**Figure 7. Mfn1[S329P] is defective for GTP-dependent oligomerization.**
**(A)** Representative blue native–PAGE of mitochondria isolated from Flp-In TREx cells expressing Mfn1[WT]-FLAG or Mfn1[S329P]-FLAG after incubation with 0.2 μg/ml TET for 4 h. Mitochondria were untreated or incubated in the presence of GMP-PNP (PNP), followed by solubilization and separation by blue native–PAGE and immunoblotted with α-FLAG. Arrow indicates ~200-kD species, closed arrowhead indicates ~320-kD species, and open arrowhead indicates ~450-kD species. Molecular-weight markers are indicated in kD on left. **(A, B)** Quantification of native mitofusin species indicated in A. Error bars represent mean ± SEM from n = 3 separate experiments.

expressing a cytosolic variant of mitofusin. Under these conditions, mitochondrially anchored mitofusin would interact with cytosolic mitofusin rather than engaging in *trans* complexes, which would prevent the excessive mitochondrial tethering and perinuclear collapse (Fig 8A). To do this, we expressed a mitofusin variant shown to have primarily cytosolic localization, Mfn1[F646D] (Huang et al, 2017). Of note, when Mfn1[F646D] was observed in the mitochondrial outer membrane in cells, mitochondrial perinuclear clusters were reported by Huang et al (2017). After transfection of TREx cells with either eGFP-Mfn1[F646D], eGFP-Mfn1[F646D-S329P], or eGFP alone, ex-pression of either Mfn1[WT]-FLAG or Mfn1[S329P]-FLAG was induced, and mitochondrial structure was assessed in cells expressing cytosolic eGFP-Mfn1[F646D] (Fig 8B and C). Cells expressing Mfn1[WT]-FLAG and either eGFP-Mfn1[F646D] or eGFP-Mfn1[F646D-S329P] had a negligible proportion of cells with mitochondrial perinuclear clusters com-pared with the eGFP-only control. As expected, most cells expressing Mfn1[S329P] and eGFP had perinuclear mitochondrial clusters. In contrast, cells that expressed both Mfn1[S329P] and cy-tosolic eGFP-Mfn1[F646D] or eGFP-Mfn1[F646D-S329P] had significantly less mitochondrial clustering than cells without cytosolic mitofusin. The difference in the percent of cells with mitochondrial perinuclear clusters between eGFP-Mfn1[F646D-S329P] and eGFP-Mfn1[F646D] could be because of trace amounts of mitofusin localizing to mitochondria in cells expressing eGFP-Mfn1[F646D-S329P]. Our data indicate that peri-nuclear clustering of mitochondria that possess Mfn1[S329P] is pri-marily the result of mitofusin interactions in *trans*, resulting in mitochondrial tethering and perinuclear collapse of the network.

# Discussion

Taken together, our data demonstrate that amino acid substitu-tions in Hinge 2 of either mitofusin paralog can result in extensive mitochondrial tethering because of prolific and stable inter-action of mitofusins on separate mitochondria. As a result, the

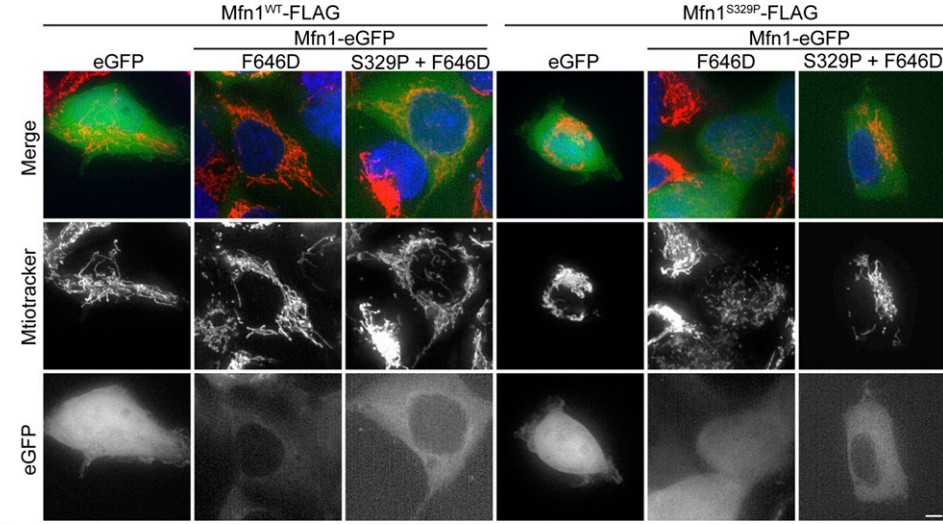

**Figure 8. Cytosolic mitofusin blocks mitochondrial tethering by mitofusin proline variants.**
**(A)** Schematic of mitochondria tethered by mitofusin *trans* complexes (left) and mitochondria with mitofusin *trans* complexes blocked by interaction with the cytosolic variant of mitofusin (F646D) (right). **(B)** Representative images of Flp-In TREx HEK293 transfected with eGFP, eGFP-Mfn1^F646D, or eGFP-Mfn1^F646D-S329P and expressing either Mfn1^WT-FLAG or Mfn1^S329P-FLAG after incubation with 0.2 μg/ml TET for 4 h. Mitochondria were labeled with MitoTracker Red CMXRos, and nuclei were labeled with NucBlue and visualized by live-cell fluorescence microscopy. Images represent maximum intensity projections. Scale bar = 5 μm. **(B, C)** Quantification of the mitochondrial morphology in the cell lines described in (B). Error bars represent mean ± SEM from n = 3 separate blinded experiments (≥100 cells per experiment).

mitochondrial network collapses to the perinuclear space. This change in mitochondrial distribution does not require dynein-dependent transport. In fact, we observed more cells with a perinuclear mitochondrial network when microtubules were depolymerized, suggesting that attachment to microtubules may slow the collapse of the mitochondrial network to the perinuclear space. Previous studies have indicated that some CMT2A variants, including Mfn2^R94Q, reduce mitochondrial transport in axons by dynein and kinesin (Misko et al, 2010, 2012). Our data indicate that aberrant mitochondrial tethering could be another factor contributing to dysfunction of these variants.

The change in mitochondrial distribution evoked by expression of the mitofusin Hinge 2 variants requires normal catalytic activity including GTP binding and formation of the intermolecular GTPase domain interface. For Mfn1, expression of either Mfn1^I105A or Mfn1^S329P resulted in a significant proportion of cells with mitochondria in the perinuclear space. Both can form the

intermolecular GTPase interface, but only Mfn1^I105A has severely diminished GTPase activity (Yan et al, 2018). For Mfn1^S329P, interfering with the formation of the G–G interface with the E209A or H107A substitutions abolished the mitochondrial perinuclear clusters, and impairing GTPase activity with the I105A substitution reduced Mfn1^S329P-induced mitochondrial clusters. Together, this suggests that mitochondrial tethering is mediated by intermolecular interactions via the GTPase domain, consistent with previous models, although our data suggest that GTP hydrolysis is not required (Cao et al, 2017; Yan et al, 2018). We postulate that GTP hydrolysis triggers conformational changes in Hinge 2, which are defective for Mfn1^S329P, and this results in the more severe mitochondrial clustering phenotype for Mfn1^S329P compared to Mfn1^I105A. Structural analysis indicates that Mfn2 has distinct catalytic mechanisms (Li et al, 2019), which is consistent with our data. It is not clear why Mfn2-null cells expressing either Mfn2^E230A or Mfn2^H128A possess mitochondrial perinuclear clusters, but either of

these substitutions reduces the effect of Mfn2$^{S350P}$, which suggests that the mechanisms may be different. Despite these functional distinctions, our analyses indicate that Mfn1$^{S329P}$ and Mfn2$^{S350P}$ both induce mitochondrial perinuclear clusters by the formation of stable intermitochondrial tethers.

For membrane division DSPs such as dynamin and DRP1, GTP hydrolysis is required to remodel the helical protein assembly, contributing to the power stroke that constricts the target membrane, and to disassembly of the DSP oligomer (Antonny et al, 2016; Ford & Chappie, 2019). Consistent with this, our data indicate that mitofusin GTP hydrolysis is required after establishment of the *trans* complex, perhaps as part of the power stroke that would drive membranes together to induce lipid mixing. In contrast, previous models suggested that mitofusin tethering occurred after GTP hydrolysis as liposomes decorated with Mfn1$_{IM}$ required the transition-state mimic GDP·BeF$_3^-$ to form proteoliposome clusters (Cao et al, 2017). This difference is likely because our analysis is with full-length protein in its native environment rather than the truncated mini-mitofusin. Our model for mitofusin-mediated membrane fusion is distinct from atlastin, a DSP responsible for ER membrane fusion. One model for ER membrane fusion suggests that GTP hydrolysis leads to *trans* interactions across ER membranes (Byrnes et al, 2013; Liu et al, 2015; O'Donnell et al, 2017), whereas other data indicate that GTP hydrolysis by Atlastin is required for disassembly of the crossover dimer after membrane fusion (Winsor et al, 2018; Crosby & Lee, 2022).

Membrane fusion occurs through regulated assembly of dynamic protein complexes that change in tertiary and quaternary conformation. We have shown that mitofusin exists primarily as a dimer in the mitochondrial outer membrane, which likely corresponds to the diffuse pattern we observe with mitofusin-mNeonGreen expression. We propose that this apo-state *cis* dimer is in a low energy state and does not engage in *trans* complex formation. Our data suggest that GTP binding switches mitofusin to a state that is fusion competent and engages in *trans* complexes to tether mitochondria via the G–G interface. The fact that amino acid substitutions in Hinge 2 prevent progression from membrane tethering to membrane fusion indicates that Hinge 2 functions in this step. Two distinct structures of mini-mitofusin constructs indicate that Hinge 2 participates in a large conformational change from an open to a closed state. The proline substitution in Hinge 2 may interfere with these conformational dynamics after GTP hydrolysis in the tethering complex, thus preventing lipid mixing and trapping mitofusin in the tethering complex. The importance Hinge 2 conformational dynamics is conserved as the yeast outer membrane fusion protein, Fzo1, also requires this domain for fusion activity (De Vecchis et al, 2017; Anton et al, 2019). Specifically, amino acid substitutions that disrupt the salt–bridge interactions between the GTPase domain and HB1 in the closed state abolish Fzo1-mediated fusion.

We have previously characterized other CMT2A-associated mitofusin variants with amino acid substitutions in either the GTPase domain or within the predicted Hinge 1 domain that connects HB1 to a second helical bundle (HB2). All of these had reduced GTP-dependent oligomerization (Engelhart & Hoppins, 2019; Samanas et al, 2020). Here, we report that the proline substitution at S329 in Hinge 2 also prohibits *cis* oligomerization but

does not alter the steady state dimer. Given that these substitutions are found in three different structural domains of mitofusin, it is unlikely that all substitutions alter a single assembly interface. Therefore, our data suggest that GTP binding evokes allosteric changes in multiple mitofusin domains to expose *cis* oligomerization interfaces. Defects in mitofusin *cis* assembly correlate with decreased mitochondrial fusion activity (Engelhart & Hoppins, 2019; Samanas et al, 2020). Consistent with this, Mfn1$^{S329P}$ and Mfn2$^{S350P}$ lack fusion activity (Fig 3). Unlike our previously characterized CMT2A variants, which were only loss of function, Mfn1$^{S329P}$ and Mfn2$^{S350P}$ are loss of function and dominant negative (Fig 3C and D). This dominant negative effect is likely because of engaging wild-type mitofusin in a nonfunctional complex either in *cis* or in *trans*. When assembled as the steady state dimer with a wild-type mitofusin, the proline variant could block GTP-dependent *cis*-oligomerization. If the proline variant assembled with wild-type mitofusin in *trans*, the proline variant would block progression from tethering to membrane fusion.

Consistent with the rapid mitochondrial tethering we observed hours after expression of the proline variants, it has been previously reported that inter-mitochondrial contact events occur with relatively high frequency compared with mitochondrial fusion or division events (Wong et al, 2019). From live-cell imaging analysis, Wong et al (2019) predicted that up to 30% of mitochondria in the cells were engaged in inter-mitochondrial contacts and the vast majority of these separated and moved away from each other. It is possible that when either Mfn1$^{S329P}$ or Mfn2$^{S350P}$ are present on the mitochondrial outer membrane, these contacts permit initiation of irreversible *trans* complexes that tether mitochondria together.

# Materials and Methods

### Plasmids and cloning

The following plasmids were used in this study: pBABE-puro (#1764; Addgene), mito-PAGFP (#23348; Addgene), peGFP-N1 (Clontech), Sec61β-GFP (gift from Jodi Nunnari), pLKO.1-TRC (#10878; Addgene), psPax2 (#12260; Addgene), and VSV-G (#8454; Addgene). To create constructs containing mitofusin variants, mitofusin was cloned using Gibson assembly as previously described (Sloat et al, 2019). Variants were created using site-directed mutagenesis using Gibson assembly. After digestion with DpnI to remove template DNA, the amplified DNA was transformed into Endura or DH5-α *Escherichia coli* cells and plasmids were purified from selected colonies. All plasmids were confirmed by sequence analysis.

### Cell culture and transfection

All cells were grown at 37°C and 5% CO$_2$ and cultured in DMEM (Thermo Fisher Scientific) containing 1× GlutaMAX (Thermo Fisher Scientific) with 10% FBS (Seradigm). MEFs cells (Mfn wild-type, Mfn1-null, and Mfn2-null) were purchased from the American Type Culture Collection. Flp-In TREx host cells (Invitrogen) were a gift from Nancy Maizels. Flp-In TREx cells were maintained in media

containing 15 µg/ml blasticidin (Life Technologies) and 100 µg/ml zeocin (Invitrogen). Cells were tested for mycoplasma contamination by 4′,6-diamidino-2-phenylindole staining before each experiment.

Flp-In TREx HEK293 clonal populations were transfected with 1.5 µg peGFP-N1, 200 µl JetPRIME buffer, and 4 µl JetPRIME Transfection Reagent (Polyplus Transfection) for 4–6 h. Experiments were performed 24 h after transfection.

### Retroviral transduction

Plat-E cells (Cell Biolabs) were maintained in complete media supplemented with 1 µg/ml puromycin and 10 µg/ml blasticidin and plated at ~80% confluency the day before transfection. 350,000 Plat-E cells were plated in a six-well dish and the following day were transfected with pBABE plasmids (3 µg pBABE Mfn1 or an empty vector; 1.5–3 µg pBABE Mfn2; 3 µg mito-paGFP) using FuGENE HD (Promega) and Opti-MEM reduced serum media (Gibco). Transfection reagent was incubated overnight before a media change. Viral supernatants were collected at ~48 and 72 h post-transfection and incubated with MEFs in the presence of 8 µg/ml polybrene. ~24 h after the last viral transduction, MEF cells were split and selection was added (1 µg/ml puromycin or 200 µg/ml hygromycin B).

### Flp-In TREx clonal populations

300,000 Flp-In TREx HEK293 host cells (Thermo Fisher Scientific) were seeded in a six-well dish. 2 d later, cells were co-transfected with 0.2 µg pFTSH (gift from Nancy Maizels) containing gene of interest and 1.8 µg pOG44 (Life Technologies) using 4 µl JetPRIME Transfection Reagent (Polyplus Transfection) for 4–6 h. Cells were moved to two 10-cm dishes and allowed to recover for 2 d before adding selection to the media (200 µg/ml hygromycin B). Under selection, cells survived at low density, and clones were collected onto sterile filter paper dots soaked in trypsin. After expansion, whole-cell extracts from clonal populations incubated with TET were screened by Western blot analysis for mitofusin against wild-type controls.

### shRNA lentivirus

shRNA target sequences were obtained using the RNAi Consortium (TRC; Broad Institute). Oligonucleotides for shRNA against DHC (target sequence: 5′-CCCGTGATTGATGCAGATAAA-3′) were purchased from Integrated DNA Technologies, annealed and ligated into pLKO.1 - TRC viral plasmid. shRNA against LacZ in pLKO.1 - TRC (target sequence: 5′-CGCGCCTTTCGGCGGTGAAAT-3′) was a gift from Yasemin Sancak. One million HEK293T cells were seeded in 6-cm plates with 5 ml media. The next day, cells were transfected with 900 ng psPax2, 100 ng VSV-G, and 1,000 ng viral plasmid in JetPRIME Transfection Reagent with JetPRIME buffer for 48 h. Viral supernatant was collected, passed through a 0.45-µm PES membrane filter, and stored at −80°C. Flp-In TREx stable cell lines were seeded at 250,000 cells in a six-well dish. The next day, viral supernatant was thawed at room temperature was added to cells with 8 µg/ml polybrene and incubated for 48 h. Cells were moved to a 10-cm dish, and selection was added (1 µg/ml puromycin). ~3 d later, cells were

seeded onto no. 1.5 glass-bottomed dishes (MatTek) coated with 10 µg/ml fibronectin (Sigma-Aldrich) and were imaged 2 d later after incubation with 0.2 µg/ml tetracycline hydrochloride (Thermo Fisher Scientific) as described below.

### Microscopy

All cells were plated ingrown on no. 1.5 glass-bottomed dishes (MatTek) for ~48 h before imaging. Cells were incubated with 0.1 µg/ml Mito-Tracker Red CMXRos with or without three drops NucBlue (Molecular Probes) for 15 min at 37°C with 5% $CO_2$. After this, MEF cells were rinsed into complete media for at least 45 min before imaging, and the Flp-In TREx cells were incubated with 0.2 µg/ml tetracycline hydrochloride (Thermo Fisher Scientific) for 4 h. A Z-series with a step size of 0.3 µm was collected with a Nikon Ti-E wide-field microscope with a 60 × NA (numerical aperture) 1.4 oil objective (Nikon), a solid-state light source (Spectra X; Lumencor), and an sCMOS camera (Zyla 5.5 megapixel). Each cell line was imaged on at least three separate occasions by a blinded experimenter (n > 100 cells per experiment).

### Image analysis

Images were deconvolved using 15 iterations of 3D Landweber deconvolution. Deconvolved images were then analyzed using Nikon Elements software. Maximum intensity projections were created using ImageJ software (NIH). Mitochondrial morphology in mammalian cells was scored as follows: reticular indicates that fewer than 50% of the mitochondria in the cell were fragments (fragments defined as mitochondria less than 2.5 µm in length); fragmented indicates that most of the mitochondria in the cell were less than 2.5 µm in length; clustered indicates that the mito-chondrial distribution was altered such that most mitochondria were coalesced in the perinuclear space. In MEF cells expressing mitofusin-mNeonGreen, only cells with GFP signal were scored. In MEF cells expressing Mfn2-mNeonGreen as described in Fig 6, only cells with GFP signal less than 1,000 units were scored. All quantifications were performed by a blinded experimenter.

### Photoactivatable mitochondrial (mt)–GFP

Cells transduced with mito-paGFP (#23348; Addgene) and either Mfn1-FLAG pBabe-hygro or Mfn2-FLAG-P2A-blasticidin pBabe were plated in No. 1.5 glass-bottomed dishes (MatTek). MEFs were in-cubated with 0.1 µg/ml MitoTracker Red CMXRos for 15 min at 37°C with 5% $CO_2$, washed, and incubated with complete media for at least 45 min before imaging. MEFs were imaged at 37°C with 5% $CO_2$. A region that was ~1 $µm^2$ was activated using a 405-nm laser, and the same cell was imaged every 10 min for 50 min. Images were collected with a Nikon Ti-E widefield microscope with a 63× NA 1.4 oil objective (Nikon), a solid-state light source (SPECTRA X; Lumencor), and an sCMOS camera (Zyla 5.5 Megapixel). Images were cropped and deconvolved using 15 iterations of 3D Landweber deconvolution on Offline Deconvolution software (Nikon). Data were then analyzed using Nikon NIS-Elements Analysis software. Background was removed, and maximum intensity projections were created. Each channel was thresholded separately for each cell, and the number of pixels that were both mCherry positive and GFP

positive were recorded. Spread of paGFP was calculated as the number of pixels that were both GFP and mCherry positive divided by total number of mCherry pixels after 50 min divided by the same variable at 0 min (immediately after activation). Graphs were created in Prism (GraphPad).

## SDS–PAGE, Western blot analysis, and quantification

Whole-cell protein lysates were obtained by resuspending PBS-washed cells in radioimmunoprecipitation assay (RIPA) lysis buffer (150 mM NaCl, 1% Nonidet P-40, 1% sodium deoxycholate, 0.1% SDS, 25 mM Tris [pH 7.4], and 1× Halt Protease Inhibitor Cocktail, EDTA-free [Thermo Fisher Scientific]). The samples were incubated on ice for 5 min and then spun at 21,000$g$ for 15 min at 4°C. The supernatant was transferred to a clean tube, and protein concentration was measured by bicinchoninic acid assay (Thermo Fisher Scientific). After separation by SDS–PAGE, proteins transferred to nitrocellulose were detected using primary rabbit or mouse antibodies and visualized with appropriate secondary antibodies conjugated to IRDye (Thermo Fisher Scientific). The following antibodies were used in this study: mouse monoclonal anti-FLAG (Sigma-Aldrich; 1:1,000); mouse monoclonal anti-Mfn2 (Sigma-Aldrich clone 4H8; 1:1,000); mouse monoclonal anti-α tubulin (Thermo Fisher Scientific clone DM1A; 1:5,000); rabbit polyclonal anti-Mfn1 (gift from Jodi Nunnari, University of California, Davis; 1:500); and rabbit polyclonal anti-DYNC1H1 (Proteintech; 1:1,000). Western blot images for expression of mitofusin in FLP-In TREx HEK293 cells were imaged on a LI-COR Imaging System (LI-COR Biosciences). Mitofusin band intensity was normalized to tubulin band intensity and compared with the vector control in the presence of TET using LI-COR Analysis Software (LI-COR Biosciences). Western blot images for shRNA were acquired on Sapphire, and quantification was performed with Azure-eSpot. Knockdown quantification was normalized using whole-protein stain (Azure Biosystems), and the mean was calculated from three separate Western blots of three independent shRNA experiments.

## Electron microscopy

Cells were pelleted and fixed in 4% glutaraldehyde and 0.1 M sodium cacodylate buffer, pH 7.3, at room temperature overnight. Samples were imaged using the JEOL 1230 transmission electron microscope (JEOL USA) operated at 80KV.

## Preparation of mitochondria

For each experiment, three 15-cm plates of HEK293 Flp-In TREx cells were grown to ~90% confluency. Expression of mitofusin was induced by incubation with 0.2 $\mu$g/ml tetracycline hydrochloride (Thermo Fisher Scientific) for 4 h. Cells were harvested by cell scraping, pelleted, and washed in mitochondrial isolation buffer (MIB) (0.2 M sucrose, 10 mM Tris-MOPS, pH 7.4, 1 mM ethylene glycol-bis($\beta$-aminoethyl ether)-N,N,N',N'-tetraacetic acid). The cell pellet was resuspended in one cell pellet volume of cold MIB with 1× HALT protease inhibitor cocktail (Thermo Fisher Scientific) and 0.5 mM phenylmethylsulfonyl fluoride (Sigma-Aldrich), and cells were homogenized by ~12 strokes on ice with a Kontes-Potter-Elvehjem tissue grinder set at 350 rpm. The homogenate was centrifuged (400$g$, 5 min, 4°C) to remove nuclei and unbroken cells, and homogenization of the pellet fraction was repeated

followed by centrifugation at 400$g$, 5 min, 4°C. The supernatant fractions were combined and centrifuged (7,400$g$, 8 min, 4°C) to pellet a crude mitochondrial fraction. The crude mitochondrial pellet was resuspended in a small volume of MIB. Protein concentration of fractions was determined by Bradford assay (Bio-Rad Laboratories) and adjusted to 5 $\mu$g/$\mu$l.

## Blue native–PAGE

Isolated mitochondria (12.5 $\mu$g) were incubated with or without 2 mM GMP-PNP (guanosine 5'-$\beta$,$\gamma$-imidotriphosphate trisodium salt hydrate; Sigma-Aldrich) as indicated at 37°C for 30 min. Mitochondria were then lysed in 1% wt/vol digitonin, 50 mM Bis-Tris, 50 mM NaCl, 10% wt/vol glycerol, 0.001% Ponceau S, pH 7.2, for 15 min on ice. Lysates were centrifuged at 16,000$g$ at 4°C for 30 min. The cleared lysate was mixed with Invitrogen NativePAGE 5% G-250 sample additive to a final concentration of 0.25%. Samples were separated on a Novex NativePAGE 4–16% Bis-Tris Protein Gels (Invitrogen) at 4°C. Gels were run at 40 V for 30 min and then 100 V for 45–60 min with dark cathode buffer (1× NativePAGE Running Buffer [Invitrogen], 0.02% [wt/vol] Coomassie G-250). Dark cathode buffer was replaced with light cathode buffer (1× NativePAGE Running Buffer [Invitrogen], 0.002% [wt/vol] Coomassie G-250), and the gel was run at 250 V for 60–75 min until the dye front ran off the gel. After electrophoresis was complete, gels were transferred to polyvinylidene fluoride membrane (Bio-Rad Laboratories) at 30 V for 16 h in transfer buffer (25 mM Tris, 192 mM glycine, 20% methanol). Membranes were incubated with 8% acetic acid for 15 min to fix the proteins to the membrane and then washed with dH$_2$O for 5 min. Membranes were dried at room temperature for 60 min and then rehydrated in 100% methanol and washed in dH$_2$O. Membranes were blocked in 4% milk for 20 min and were probed with anti-FLAG (Sigma-Aldrich) overnight at 4°C. Membranes were incubated with the HRP-linked secondary antibody (Cell Signaling Technology) at room temperature for 1 h. Membranes were developed in Radiance Plus Chemiluminescent HRP Substrate (Azure Biosystems) for 5 min and imaged on a Sapphire Biomolecular Imager (Azure Biosystems). Band intensities were quantified using AzureSpot analysis software (Azure Biosystems). NativeMark Unstained Protein Standard (Life Technologies) was used to estimate molecular weights of mitofusin protein complexes.

## Protein expression and purification Mfn1$_{IM}$

The Mfn1$_{IM}$ pET28 plasmid was obtained from Song Gao. Mutations were made by Gibson assembly and confirmed by sequencing. Mfn1$_{IM}$ constructs were expressed in *E. coli* Rosetta (DE3) cells. Cells were cultured in Luria-Bertani medium with 150 $\mu$g/ml ampicillin and 25 $\mu$g/ml chloramphenicol at 37°C to an OD$_{600}$ of ~0.6, and protein expression was induced by the addition of 100 $\mu$M isopropyl 1-thio-$\beta$-D-galactopyranoside. Induced cultures were grown overnight at ~17°C. The cells were harvested by centrifugation at 6,000$g$ for 10 min. Cell pellets expressing MFN1$_{IM}$ were resuspended in 5 ml of PBS, pelleted by centrifugation at 6,000$g$ for 5 min, frozen in liquid nitrogen, and stored at –80°C. Cells were thawed in a room-temperature water bath and resuspended in 50 ml of 50 mM Hepes-KOH (pH 7.4), 400 mM NaCl, 5 mM MgCl$_2$, 30 mM imidazole, 1 mM

phenylmethanesulfonylfluoride, 1× protease inhibitor mixture (Thermo Fisher Scientific), and 2.5 mM $\beta$-mercaptoethanol ($\beta$-ME) and lysed using a microfluidizer (Avestin). The lysate was subjected to centrifugation at 34,000$g$ for 45 min. The supernatant was applied to 2.5 ml of HisPurTM Ni-NTA beads (Thermo Fisher Scientific) equilibrated with binding buffer 1 (20 mM Hepes-KOH [pH 7.4], 400 mM NaCl, 5 mM $MgCl_2$, 30 mM imidazole [pH 8.0], and 2.5 mM $\beta$-ME) and nutated at 4°C for 30 min. Ni-NTA beads bound to protein were washed with 20 column volumes of binding buffer 1, and proteins were eluted with elution buffer (20 mM Hepes-KOH [pH 7.4], 400 mM NaCl, 5 mM $MgCl_2$, 300 mM imidazole, and 2.5 mM $\beta$-ME). $Mfn1_{IM}$-containing elutions were incubated with 800 $\mu$g of GSHS transferase (GST)–fused PreScission protease to remove the N-terminal His6 tag. This was dialyzed overnight against binding buffer 2 (20 mM Hepes-KOH [pH 7.4], 400 mM NaCl, 5 mM $MgCl_2$, and 2.5 mM $\beta$-ME). After dialysis, PreScission protease was removed using a GST column. The protein was reapplied to a second Ni-NTA column equilibrated with binding buffer 2. Binding buffer 1 was used to elute the protein, which was subsequently loaded onto a Superdex200 16/60 column (GE Healthcare) equilibrated with gel filtration buffer (20 mM Hepes-KOH [pH 7.4], 150 mM NaCl, 5 mM $MgCl_2$, and 1 mM DTT). The protein eluted in a discrete peak corresponding to a molecular mass of ~50 kD. Protein was concentrated on an Amicon Ultra Centrifugal Filter (molecular-weight cutoff, 30,000) (Millipore) to 30 mg/ml, and glycerol was added to 20% before the protein was aliquoted and stored at –80°C. Protein purification was performed at 4°C. Protein concentration was determined by Bradford assay (Bio-Rad).

### GTPase assay

Frozen protein was thawed on ice and diluted to 2.5 $\mu$M in 20 mM Hepes-KOH (pH 7.4), 50 mM KCl, 5 mM $MgCl_2$, and 1 mM DTT. Protein concentrations were confirmed by Bradford assay (Bio-Rad). Reactions were set up in triplicate in a 96-well plate on ice. Variable concentrations of GTP were added, and reactions were incubated at 37°C for 15 min. Reactions were stopped by adding 200 mM EDTA on ice. Concentrations of free inorganic phosphate (Pi) were measured by malachite green reagent, incubated at room temperature for 15 min. The optical density at 650 nm was measured, and a potassium phosphate standard curve was used to determine release of Pi by GTP hydrolysis. Data were analyzed in Prism (GraphPad).

### Sucrose gradient ultracentrifugation

Linear gradients (5–20%) were generated by layering 2.5 ml of 5% sucrose buffer (20 mM Tris [pH 8], 150 mM KCl, 4 mM $MgCl_2$, and 5% sucrose) on top of 2.5 ml 20% sucrose buffer (20 mM Tris [pH 8], 150 mM KCl, 4 mM $MgCl_2$, and 20% sucrose) in an ultracentrifuge tube. The tube was parafilmed and laid horizontally for 2 h at room temperature then vertically for 1 h at 4°C. 150 $\mu$g purified protein or protein standards (HMW calibration kit; GE Healthcare) were pipetted on top of the gradient. If nucleotide was included in the experiment, it was added at a final concentration of 2 mM GDP, with or without 2.5 mM $BeSO_4$ and 25 mM NaF, and these were incubated for 10 min on ice followed by 30 min at 37°C and finally 15 min on ice before loading onto the gradient. Gradients were spun at 100,000$g$ for 16 h at 4°C. Fractions of 250 $\mu$l were collected and subject to SDS–PAGE and Coomassie staining.

### Immunofluorescence

Cells were plated onto fibronectin-coated coverslips. Cells were incubated with 0.1 $\mu$g/ml MitoTracker Red CMXRos for 15 min at 37°C with 5% $CO_2$, then washed into compete media containing 1 $\mu$g/ml tetracycline hydrochloride (Thermo Fisher Scientific) ± 5 nM nocodazole (Thermo Fisher Scientific) and incubated for 4 h. Cells were fixed in 4% paraformaldehyde (Sigma-Aldrich) and 0.5% glutaraldehyde (Polysciences Inc.) for 15 min at room temperature and then solubilized in 0.1% Surfact-Amps TX-100/PBS (Thermo Fisher Scientific). Microtubules were stained using mouse monoclonal anti-$\alpha$ tubulin (Thermo Fisher Scientific clone DM1A; 1:300) and the goat anti-mouse Alexa 488 secondary antibody (Invitrogen; 1:300).

## Supplementary Information

## Acknowledgements

We thank members of the Hoppins lab and Marijn Ford for scientific suggestions and critical reading of the manuscript. Funding was provided by the Department of Biochemistry, University of Washington, and through grants from the National Institute of General Medical Sciences (Grant No. R01GM118509 to S Hoppins).

### Author Contributions

SR Sloat: conceptualization, data curation, formal analysis, methodology, and writing—original draft.
S Hoppins: conceptualization, data curation, formal analysis, supervision, funding acquisition, investigation, methodology, project administration, and writing—original draft, review, and editing.

### Conflict of Interest Statement

The authors declare that they have no conflict of interest.

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
