## [Reviewer comments · Life Science Alliance]

Life Science Alliance

A dominant negative mitofusin causes mitochondrial perinuclear clusters due to aberrant tethering

Stephanie Sloat and Suzanne Hoppins

DOI: <https://doi.org/10.26508/lsa.202101305>

Corresponding author(s): *Suzanne Hoppins, University of Washington*

Review Timeline:

Submission Date:	2021-11-19
Editorial Decision:	2021-12-20
Revision Received:	2022-08-05
Editorial Decision:	2022-08-30
Revision Received:	2022-09-27
Accepted:	2022-09-28

Scientific Editor: Novella Guidi

Transaction Report:

December 20, 2021

Re: Life Science Alliance manuscript #LSA-2021-01305-T

Dr. Suzanne Hoppins
University of Washington
1959 NE Pacific St Health Sciences Building J383
Seattle, WA 98195

Dear Dr. Hoppins,

Thank you for submitting your manuscript entitled "A dominant negative mitofusin variant reveals that mitochondrial tethering requires GTP hydrolysis" to Life Science Alliance. The manuscript was assessed by expert reviewers, whose comments are appended to this letter. We, thus, encourage you to submit a revised version of the manuscript back to LSA that responds to all of the reviewers' points.

Thank you for this interesting contribution to Life Science Alliance. We are looking forward to receiving your revised manuscript.

Sincerely,

B. MANUSCRIPT ORGANIZATION AND FORMATTING:

Reviewer #1 (Comments to the Authors (Required)):

This study by Sloat & Hoppins characterizes a CMT2A disease-associated Ser to Pro missense mutation in hinge 2 of Mfn2 (S350P) and the equivalent mutation in Mfn1 (S329P). Prior to this study it was known that both Mfn1 and Mfn2 paralogs are required for efficient mitochondrial membrane fusion; Mfn1/2 undergoes G-G dimerization and a large rotational conformation change by HB1 about hinge 2 coupled to GTP binding and/or hydrolysis for membrane tethering and fusion; Mfn oligomerizes both in cis and in trans and both modes are required for fusion; trans complex formation mediated by G-G dimerization likely functions as a tether to link adjacent membranes; and G-G dimerization is a prerequisite for GTP hydrolysis.

In this study, the authors use cell-based mitochondrial morphological assays to show that 1) the S329P/S350P mutation in either Mfn1 or Mfn2 fails to restore normal reticular mitochondrial morphology to Mfn1 or Mfn2 knockout cells, respectively, indicating a lack of fusion activity; 2) the mutation instead causes massive perinuclear clustering of mitochondria that appear relatively normal as far as mitotracker labeling and EM ultrastructure are concerned; 3) the perinuclear clustered mitochondria remain fragmented consistent with lack of fusion; and 4) clustering does not depend on microtubule-based motility as assessed by dynein knockdown. Also, the GTP hydrolysis rate for the hinge mutant does not differ substantially from the WT. These findings suggest that the S329P/S350P mutation allows tethering of mitochondria in trans but not their subsequent fusion, causing the perinuclear clustering phenotype.

From here, using the perinuclear clustering morphological assay as a readout for tethering, the authors take advantage of several previously characterized catalytic mutations in the Mfn1/2 G domain to probe whether tether formation depends on GTP binding, G-G dimerization and/or GTP hydrolysis. To test the role of GTP binding, they use the W239A/W260A mutation shown to block GTP binding (Cao et al, 2017). To test the role of GTP hydrolysis, they use the H107A/H128A mutation shown to block GTP hydrolysis but not GTP binding (Cao et al, 2017). Finally, to test the role of G-G dimerization, they use E209A and R238A (or E230A and R259A) shown to both disrupt the G-G interface and GTP hydrolysis (Cao et al, 2017). The results are stronger for Mfn1 than Mfn2, but the trend is similar: All three perturbations to the GTPase cycle disrupt tethering. From this, the authors conclude that tethering requires all three, GTP binding, G-G dimerization and GTP hydrolysis.

Overall, this study is methodical and well-described. The morphological data and their quantification are generally convincing. However, the main conclusion of the paper indicated by the title, that tethering requires GTP hydrolysis, is an overinterpretation of the data. This is because the conclusion rests heavily on the H107A mutant, which, as acknowledged in passing by the authors, may have multiple effects in the GTPase cycle. Supporting a complicated role(s) for H107, it has a different orientation in different crystal structures. For example, in one structure, it points toward the active site consistent with a crucial role in hydrolysis. But in other structures, it points away from the active site. Indeed, in a GTP-bound structure (5gof), H107 points away from the active site in a manner consistent with its role in mediating G-G dimerization (as implied by the model in Fig 4e in Cao et al, 2017). This would be analogous to the similarly complicated role of R77 in atlastin-1, analogous to H107, which not only stabilizes the transition state of hydrolysis, but also mediates G-G dimer contact. Therefore, it is plausible that H107A impedes tethering in the authors' assay not so much by preventing GTP hydrolysis, but by weakening the G-G interface, which is in turn required for hydrolysis. Two other mutations analyzed in this study E209A and R238A also disrupt the G-G interface and indirectly block GTP hydrolysis (Cao et al, 2017). Considering this major caveat, it is difficult to conclude that tethering requires GTP hydrolysis. Either H107A needs to be shown not to inhibit G-G dimer formation; another more selective mutation needs to be used to test the requirement for hydrolysis; or the conclusion significantly softened to take the caveats into consideration.

Minor:

1. The GTPase data in Fig 6A lacks clarity. First, the y-axis reads V (kcat/min) but should read either V ($\mu\text{M}/\text{min}$) or kcat (min^{-1}). It looks like V ($\mu\text{M}/\text{min}$) is being plotted, given the similarity to the V_{max} values in the table. Second, both the catalytic rate and KM of wild type protein in this study seem much higher than reported previously (Cao et al, 2017). This should be commented on. Third, the methods for the GTPase assay indicate that only one time point, 15 min, was measured. Typically, multiple time points are taken to assess linearity of the assay.

Reviewer #2 (Comments to the Authors (Required)):

The research paper by Sloat & Hoppins titled 'A dominant negative mitofusin variant reveals that mitochondrial tethering requires GTP hydrolysis' provides novel insight into the mechanism by which MFN2 functions. By studying the S350P variant in MFN2, which causes peripheral neuropathy, as well as the corresponding S329P variant in MFN1, they begin to untangle the role of GTPase activity in the mitochondrial tethering role of MFNs. They find that these variants cause perinuclear clustering of mitochondria due to excessive tethering that cannot progress to fusion. This clustering does not occur in other organelles, and does not require dynein-directed microtubule transport. Next, by introducing a series of additional mutations that impact GTPase hydrolysis and binding, they argue that fully functional GTPase activity is required to induce the clustering phenotype, and by default is required for tethering. Then they show that the MFN1 S329P variant leads to reduces nucleotide-dependent cis-assembly. Finally, by expressing a cytosol localized version of MFN1, which competes for MFN binding, they show that perinuclear clusters require trans assembly. While the findings are important and reveal novel mechanistic insight into how MFN2 mediates mitochondrial tethering, there are still a few issues that need to be addressed.

Major points

The authors should discuss why they choose to study the S350P variant, as opposed to any of the other >100 known variants that cause CMT2A. They should also discuss what is known about the pathology linked to this variant, and its mode of inheritance (presumably heterozygous), as this relates to their claim that the variant is dominant negative. The authors should also mention the specific variant earlier in the paper, certainly in the abstract if not in the introduction.

The authors need to discuss the limitations of published MFN structures (mini-mitofusin), which are lacking a significant portion of the MFN protein, including the transmembrane domain. This is important given recent work showing that the topology of MFNs is different than the initially proposed arrangement where both the N- and C- terminus are exposed to the cytosol (PMID: 29212658). Instead, the C-terminus is located in the inner membrane space, and is thus not available to interact with the N-terminus. As such, the published MFN models, where the C-terminus interacts with the N-terminus, are not likely to be completely relevant. That being said, I think that there is still some structural information that is likely relevant and pertinent to this study, with respect to the putative hinge 2 domain that may not depend on the C-terminal domain being present.

Another issue related to the topology of MFN2, is previous work showing that the addition of a C-terminal tag affects the topology of the C-terminus, such that it is exposed to the cytosol, rather than the inner membrane space (PMID: 29212658), which is the case for the endogenous protein. Thus, it is possible that the constructs used in this study, many of which have C-terminal tags, do not have the same topology of endogenous MFNs, and this could be a confounding issue. To address this issue, the authors should show that an untagged version of MFN2 S350P also causes perinuclear clustering.

Minor issues

The authors rightly discuss the potential issues related to overexpressing MFN2 and perinuclear clumping, which has been described previously. Although they show that the expression level of WT and mutant MFNs is comparable in their studies with Tet-induced expression, it would also be nice to see equal expression levels in their original overexpression studies corresponding to figure 2.

For the TET induced expression, they may not want to use the term 'rapid' in the context of clustering given that fusion and tethering events occur on a timescale of seconds to minutes, and they are looking at a single 4-hour timepoint. Is this the earliest that they can detect protein expression?

It was a bit curious that only the MFN1 variants were examined in Figures 7 & 8, and that nothing was done for the MFN2 variants. Is there any specific reason for this, or is it just that the MFN1 and MFN2 variants behaved primarily the same way in earlier experiments?

I'm not sure that I agree with the title stating that the 'variant reveals that mitochondrial tethering requires GTP hydrolysis', as it implies the variant does not have GTPase activity or tethering abilities. In fact, their data show the opposite, that the variant has GTPase activity and it leads to excessive tethering. It is only when they introduce additional mutations that impair GTPase activity that the excessive tethering is inhibited. Maybe something along the lines of 'Characterization of the S350P MFN2 variant helps to reveal that mitochondrial tethering requires GTP hydrolysis'.

Reviewer #3 (Comments to the Authors (Required)):

In the present study, the authors very nicely describe a novel dominant negative role of mitofusin mutants in perinuclear clustering of mitochondria, advancing our current knowledge of mitochondrial fusion. Mitochondrial clustering depends on both GTP binding and hydrolysis but is not the result of altered mitochondrial transport events. Cytosolic mitofusins efficiently titrated mitochondrial clustering, consistent with a trans tether trapping of mitochondria that prevents further progression through the fusion cycle. Moreover, these variants -located in a critical hinge point between the N-terminally located GTPase domain and the 1st helix bundle- mimic a mutation present in Charcot-Marie-Tooth patients, thus this manuscript also provides some hints for understanding how mitofusin 2 can cause neurodegeneration. This manuscript is largely based on mitochondrial morphology analysis, inferred from biochemical and structural features of the mini-mitofusins. It would be important to support their findings by mechanistic analysis using full length mitofusins, thus allowing to place the function of the mutated residues in the fusion

cycle and to confirm the proposed model.

Main concerns:

The authors show that perinuclear clustering depends on GTP binding and hydrolysis, and on the G-G interface. It would be important to show where in the fusion cycle these mutants are blocked:

A - To support their hypothesis that the proline mutations create hyperactive mitofusin tethering variants, the authors should analyse cis and trans binding, at least for Mfn1, with isolated mitochondria and differently tagged mitofusin versions, using all combinations of the proline mutants and previously described mutants used in Fig. 6C and 6E.

B - Similarly, in vitro fusion assays with isolated mitochondria should be performed on all mutation combinations as in Fig. 6C and 6E for Mfn1.

C- As suggested by the authors, irregular mitochondrial localization of the mitofusin variants could indicate failure in complex disassembly, like previously shown for yeast mitofusin variants (Anton et. al. 2020). Thus, the authors could quantify mitofusins loci formation, and also analyse perinuclear mitochondrial clustering of their proline mutant, upon simultaneous inhibition of mitochondrial fission, to force mitochondrial tubulation.

D-In Fig 8 the authors show that expression of cytosolic eGFP-Mfn prevents mitochondrial clustering by S329P. However, if the presumption that S3329P prevents Mfn complexes disassembly is correct, one could expect to see co-localization of eGFP-Mfn to mitochondrial-located S329P, especially given its dominant negative properties. Please comment.

E-Fig. 7 does not allow concluding that only cis-oligomerization is prevented, given that mini-mfn properties do not prove that the same will occur for the S329P variant once in the full length and membrane contexts. I would suggest to either analyse if the 200 kDa form corresponds to a cis or trans dimer, or instead eliminate this figure.

Other points:

-This study shows the importance of hinge 2 for mitofusin activity. However, this hinge has been extensively studied mechanistically, mostly in yeast, from studies performed by the Escobar and Cohen groups. Thus, the previous findings should be clearly cited. Moreover, the authors should clarify how this proline might differ or act similarly to previously described functions in this hinge.

-The authors nicely show that the mitochondrial clustering observed in the proline mutant depends on GTP binding and hydrolysis and "propose that mitofusin exists primarily in a low-energy state that is inactive and that GTP hydrolysis converts mitofusin to a high-energy state that can tether mitochondria." It has been previously shown that GTP binding is necessary for mitofusin complex formation in cis and in trans, in contrast to GTP hydrolysis. Thus, it would be important to clearly discuss these differences. Further, the exact "energy state", as it has been extensively studied for viral fusion proteins, has not been shown in this study. So, perhaps the authors could clarify what they mean.

- The authors claim that "the mitochondria in cells expressing the mitofusin proline variant possessed cristae structures similar to wild-type, consistent with normal mitochondrial function". However mitochondrial function is not show/characterised. Please clarify.

Minor comment:

-"To determine enzymatic activity of Mfn1S329P, we used the internally modified Mfn1 (Mfn1IM) that consists of residues 1-365 and 696-741 connected by a flexible linker (Cao et al., 2017). We found the kinetics of GTP hydrolysis was similar between Mfn1IM WT and Mfn1IM S329P (Fig. 6, A & B)"

For easier understanding the authors could clarify here that Mfn1IM corresponds to "mini-mitofusin 1".

We would like to thank all reviewers for taking the time to offer their constructive critique of our manuscript. We have responded to the comments as outlined below. We have resubmitted an updated version of the manuscript with changes in the text highlighted in blue, as well as a clean document. We are grateful for the value and clarity that these changes have added to the manuscript.

Reviewer #1 (Comments to the Authors (Required)):

This study by Sloat & Hoppins characterizes a CMT2A disease-associated Ser to Pro missense mutation in hinge 2 of Mfn2 (S350P) and the equivalent mutation in Mfn1 (S329P). Prior to this study it was known that both Mfn1 and Mfn2 paralogs are required for efficient mitochondrial membrane fusion; Mfn1/2 undergoes G-G dimerization and a large rotational conformation change by HB1 about hinge 2 coupled to GTP binding and/or hydrolysis for membrane tethering and fusion; Mfn oligomerizes both in cis and in trans and both modes are required for fusion; trans complex formation mediated by G-G dimerization likely functions as a tether to link adjacent membranes; and G-G dimerization is a prerequisite for GTP hydrolysis.

In this study, the authors use cell-based mitochondrial morphological assays to show that 1) the S329P/S350P mutation in either Mfn1 or Mfn2 fails to restore normal reticular mitochondrial morphology to Mfn1 or Mfn2 knockout cells, respectively, indicating a lack of fusion activity; 2) the mutation instead causes massive perinuclear clustering of mitochondria that appear relatively normal as far as mitotracker labeling and EM ultrastructure are concerned; 3) the perinuclear clustered mitochondria remain fragmented consistent with lack of fusion; and 4) clustering does not depend on microtubule-based motility as assessed by dynein knockdown. Also, the GTP hydrolysis rate for the hinge mutant does not differ substantially from the WT. These findings suggest that the S329P/S350P mutation allows tethering of mitochondria in trans but not their subsequent fusion, causing the perinuclear clustering phenotype.

From here, using the perinuclear clustering morphological assay as a readout for tethering, the authors take advantage of several previously characterized catalytic mutations in the Mfn1/2 G domain to probe whether tether formation depends on GTP binding, G-G dimerization and/or GTP hydrolysis. To test the role of GTP binding, they use the W239A/W260A mutation shown to block GTP binding (Cao et al, 2017). To test the role of GTP hydrolysis, they use the H107A/H128A mutation shown to block GTP hydrolysis but not GTP binding (Cao et al, 2017). Finally, to test the role of G-G dimerization, they use E209A and R238A (or E230A and R259A) shown to both disrupt the G-G interface and GTP hydrolysis (Cao et al, 2017). The results are stronger for Mfn1 than Mfn2, but the trend is similar: All three perturbations to the GTPase cycle disrupt tethering. From this, the authors conclude that tethering requires all three, GTP binding, G-G dimerization and GTP hydrolysis.

Overall, this study is methodical and well-described. The morphological data and their quantification are generally convincing. However, the main conclusion of the paper indicated by the title, that tethering requires GTP hydrolysis, is an overinterpretation of the data. This is because the conclusion rests heavily on the H107A mutant, which, as acknowledged in passing by the authors, may have multiple effects in the GTPase cycle. Supporting a complicated role(s) for H107, it has a different orientation in different crystal structures. For example, in one structure, it points toward the active site consistent with a crucial role in hydrolysis. But in other structures, it points away from the active site. Indeed, in a GTP-bound structure (5gof), H107 points away from the active site in a manner consistent with its role in mediating G-G dimerization (as implied by the model in Fig 4e in Cao et al, 2017). This would be analogous to the similarly complicated role of R77 in atlastin-1, analogous to H107, which not only stabilizes the transition state of hydrolysis, but also mediates G-G dimer contact. Therefore, it is plausible that H107A impedes tethering in the authors' assay not so much by

preventing GTP hydrolysis, but by weakening the G-G interface, which is in turn required for hydrolysis. Two other mutations analyzed in this study E209A and R238A also disrupt the G-G interface and indirectly block GTP hydrolysis (Cao et al, 2017). Considering this major caveat, it is difficult to conclude that tethering requires GTP hydrolysis. Either H107A needs to be shown not to inhibit G-G dimer formation; another more selective mutation needs to be used to test the requirement for hydrolysis; or the conclusion significantly softened to take the caveats into consideration.

This reviewer makes an excellent point that data from different groups suggest multi-faceted roles of several amino acid side chains in the GTPase domain and that not all activities have been previously tested.

To assess dimerization of Mfn1-H107A, we purified Mfn1_{IM} with the H107A substitution. The oligomeric state was determined using sucrose gradient ultracentrifugation. Mitofusin protein was pre-incubated with or without nucleotide and then layered on top of a linear sucrose gradient (5-20%). Following ultracentrifugation, fractions were taken and subject to SDS-PAGE and Coomassie staining. While Mfn1_{IM} migrated as a monomer in the presence of GDP, it shifted to a dimer following incubation with GDP•BeF₃⁻. In contrast, Mfn1_{IM}H107A migrates as a monomer in both conditions. These data indicate that substitution of alanine at Mfn1-H107 also interferes with formation of the intermolecular GTPase domain interface. These data are now included in Supplemental Figure 8 and here for your reference.

Therefore, we have now analyzed the impact of a substitution of alanine at Mfn1-I105 on perinuclear clusters and mitochondrial tethering by Mfn1-S329P and Mfn2-S350P.

The I105A variant was analyzed by Yan et al. (2018) where they report the following: Expression of Mfn1-I105A in Mfn1-null cells failed to restore reticular mitochondrial morphology, consistent with the conclusion that this variant cannot support mitochondrial fusion. Substitution of alanine at I105 drastically reduced GTPase activity of their Mfn1MGD, but only moderately affected dimerization via the G-G interface. Therefore, we chose to utilize Mfn1-I105A/Mfn2-I126A variants to differentiate the role of the G-G interface and GTP hydrolysis in the function of Mfn1-S329P/Mfn2-S350P.

In our hands, expression of Mfn1-I105A in Mfn1-null cells did not restore mitochondrial network connectivity, indicating a lack of fusion activity. We observed fragmented mitochondrial networks in ~35% of cells and mitochondria clustered in the perinuclear space in just less than 50% of the cells. The clustering phenotype was significantly less than we observed when Mfn1-S329P was expressed in Mfn1-null cells (>80% of cells). Interestingly, Mfn1-I105A/S329P behaved similarly to Mfn1-I105A alone, which is consistent with our model that GTPase activity contributes to the proline variant-dependent mitochondrial clusters. However, the observation that Mfn1-I105A induced mitochondrial perinuclear clusters itself suggests that GTPase activity is not required for mitochondrial tethering by Mfn1.

To verify that Mfn1-I105A does not support mitochondrial fusion and to determine if it also has dominant negative activity, we analyzed the diffusion of mitochondrial paGFP (mt-paGFP) in cells expressing Mfn1-I105A. These data reveal that Mfn1-I105A did not support mitochondrial fusion in Mfn1-null cells and blocked mitochondrial fusion in wild type cells. These data are shown in Figure S9 and below for your reference.

As is true throughout our analysis, Mfn2 tells a slightly different story, likely reflecting the differences in the catalytic activity of Mfn1 and Mfn2. Specifically, expression of Mfn2-I126A in Mfn2-null cells did not result in a substantial number of cells with perinuclear mitochondrial clusters. Furthermore, the substitution of I126 for alanine was sufficient to reduce the mitochondrial clusters induced by S350P. In the context of our analysis, this supports the conclusion that Mfn2-S350P requires normal GTPase domain function for mitochondrial perinuclear cluster formation.

Overall, these data are consistent with a model where the initial formation of a mitofusin tether or trans complex requires the intermolecular GTPase interface. Given that Mfn1-I105A possesses severely diminished GTPase activity, hydrolysis is likely required to advance from tethering to membrane fusion. We postulate that the mitochondrial clustering induced by the proline variants is more abundant due to its GTPase activity. We show that GTP hydrolysis by Mfn1-S329P is not impaired, but this variant does not support membrane fusion. Therefore, we propose that dysregulation of conformational dynamics in Hinge 2 during GTP hydrolysis by Mfn1-S329P renders this variant unable to effectively progress from membrane tethering to fusion.

These changes are reflected in the text.

Minor:

1. The GTPase data in Fig 6A lacks clarity. First, the y-axis reads V (kcat/min) but should read either V ($\mu\text{M}/\text{min}$) or kcat (min^{-1}). It looks like V ($\mu\text{M}/\text{min}$) is being plotted, given the similarity to the V_{max} values in the table. Second, both the catalytic rate and K_M of wild type protein in this study seem much higher than reported previously (Cao et al, 2017). This should be commented on. Third, the methods for the GTPase assay indicate that only one time point, 15 min, was measured. Typically, multiple time points are taken to assess linearity of the assay.

The axis title has been modified. Thank you for noting this error.

Our GTPase reaction buffer includes potassium, which increases the efficiency of GTP hydrolysis as reported in Yan et al., 2018.

Because we are measuring saturation of GTPase activity by increasing GTP concentrations, we report only one time point in this figure. A single timepoint is consistent with other publications in the field from several other labs including PMID 29022874, 21368113, 34817557, and 26783363.

Reviewer #2 (Comments to the Authors (Required)):

The research paper by Sloat & Hoppins titled 'A dominant negative mitofusin variant reveals that mitochondrial tethering requires GTP hydrolysis' provides novel insight into the mechanism by which MFN2 functions. By studying the S350P variant in MFN2, which causes peripheral neuropathy, as well as the corresponding S329P variant in MFN1, they begin to untangle the role of GTPase activity in the mitochondrial tethering role of MFNs. They find that these variants cause perinuclear clustering of mitochondria due to excessive tethering that cannot progress to fusion. This clustering does not occur in other organelles, and does not require dynein-directed microtubule transport. Next, by introducing a series of additional mutations that impact GTPase hydrolysis and binding, they argue that fully functional GTPase activity is required to induce the clustering phenotype, and by default is required for tethering. Then they show that the MFN1 S329P variant leads to reduces nucleotide-dependent cis-assembly. Finally, by expressing a cytosol localized version of MFN1, which competes for MFN binding, they show that perinuclear clusters require trans assembly. While the findings are important and reveal novel mechanistic insight into how MFN2 mediates mitochondrial tethering, there are still a few issues that need to be addressed.

Major points

The authors should discuss why they choose to study the S350P variant, as opposed to any of the other >100 known variants that cause CMT2A. They should also discuss what is known about the pathology linked to this variant, and its mode of inheritance (presumably heterozygous), as this relates to their claim that the variant is dominant negative. The authors should also mention the specific variant earlier in the paper, certainly in the abstract if not in the introduction.

These modifications have been made to the text and are included here for reference.

1. The abstract now includes references to Mfn2-S350P as the CMT2A variant of interest.
2. We have added information about the pathology of Mfn2-S350P and why we chose to study it to the introduction.

“We had previously performed a functional screen of mitofusin with CMT2A amino acid substitutions at conserved positions in Mfn1 {Engelhart and Hoppins, 2019, #169906}. We expanded the screen and observed a striking change in mitochondrial distribution when either Mfn2^{S350P} or Mfn1^{S329P} was expressed in cells. The Mfn2^{S350P} mutation is inherited as a spontaneous dominant mutation and causes early onset CMT as well as brain lesions (Cho et al., 2007)(Chung et al., 2010).”

The authors need to discuss the limitations of published MFN structures (mini-mitofusin), which are lacking a significant portion of the MFN protein, including the transmembrane domain. This is important given recent work showing that the topology of MFNs is different than the initially proposed arrangement where both the N- and C- terminus are exposed to the cytosol (PMID: 29212658). Instead, the C-terminus is located in the inner membrane space, and is thus not available to interact with the N-terminus. As such, the published MFN models, where the C-terminus interacts with the N-terminus, are not likely to be completely relevant. That being said, I think that there is still some structural information that is likely relevant and pertinent to this study, with respect to the putative hinge 2 domain that may not depend on the C-terminal domain being present.

We have made changes in the text to reflect the gaps in knowledge and discrepancies that remain in the field regarding mitofusin topology and structure. They are included here for reference. We have simplified the complex literature to a single statement, as it didn't seem appropriate to explain all the different structural and topological models that have been published. We have also performed protease protection to assess the topology of mitofusin (Samanas et al, 2020). We did not observe the same protection of the C-terminal domain as observed in Mattie et al. There are differences in our approach that are likely responsible. Our experiments were performed with stable lines expressing Mfn1-FLAG at near-endogenous levels in an Mfn1-null background. We did this to exploit the specificity of the FLAG antibody to assess the localization of the C-terminal domain. In contrast, Mattie used mitochondria isolated from HEK 293 cells and commercially available antibodies to detect native protein. Both approaches have caveats, and the matter remains to be resolved.

Pg 4 "While there are several models of mitofusin topology and structure, the core catalytic features of the mitofusin GTPase domain are consistent with other members of the DSP family."

Pg 14 "These experiments were guided by functional information uncovered from atomic resolution structures of the mini-Mfn1 constructs, which have provided significant mechanistic insight into GTP binding and hydrolysis. Of note, there are paralog-specific differences and variations between individual Mfn1 structures that raise questions about the specific role(s) of some features of the GTPase domain. Therefore, we used several amino acid substitutions in the GTPase domain to disrupt the catalytic cycle at different stages..."

Another issue related to the topology of MFN2, is previous work showing that the addition of a C-terminal tag affects the topology of the C-terminus, such that it is exposed to the cytosol, rather than the inner membrane space (PMID: 29212658), which is the case for the endogenous protein. Thus, it is possible that the constructs used in this study, many of which have C-terminal tags, do not have the same topology of endogenous MFNs, and this could be a confounding issue. To address this issue, the authors should show that an untagged version of MFN2 S350P also causes perinuclear clustering.

We have expressed N-terminally tagged Mfn1-S329P in cells and quantified mitochondrial morphology. We found that ~85% of cells expressing this construct had clustered mitochondria. This is comparable to our observations with C-terminally tagged Mfn1-S329P (Figure 2B).

We use retroviral vectors (pBABE) to express mitofusin in MEFs. We find that this results in lower expression levels than transient transfection, which is preferable for mitofusin as overexpression of wild-type mitofusin can also cause perinuclear clusters. Our transduction results in a mixed population, so we chose to use the N-terminal GFP tag to address this point so that we could score only cells expressing the construct.

Minor issues

The authors rightly discuss the potential issues related to overexpressing MFN2 and perinuclear clumping, which has been described previously. Although they show that the expression level of WT and mutant MFNs is comparable in their studies with Tet-induced expression, it would also be nice to see equal expression levels in their original overexpression studies corresponding to figure 2.

We do not report this because it is misleading due to the extreme expression differences within the transduced population. These experiments generate a mixed population where many cells do not express detectable protein, despite conferring resistance to the vector markers (hyg or puro). Therefore, we score cells that have green fluorescence that falls within a range that we know to be near endogenous based on clonal populations previously generated in the lab. This process is described in the methods: "In MEF cells expressing mitofusin-mNeonGreen, only cells with GFP

signal were scored. In MEF cells expressing Mfn2-mNeonGreen as described in Figure 6, only cells with GFP signal less than 1000 units were scored. All quantification was performed by a blinded experimenter.”

For your reference we include here a Western Blot analysis of either Mfn1-null or wild-type cells expressing Mfn1-mNG. These blots demonstrate that within the mixed population of cells, the expression of exogenous mitofusin (Mfn1-mNeonGreen) is far below endogenous levels (Mfn1).

For the TET induced expression, they may not want to use the term 'rapid' in the context of clustering given that fusion and tethering events occur on a timescale of seconds to minutes, and they are looking at a single 4-hour timepoint. Is this the earliest that they can detect protein expression?

Rapid is a relative term, so we have provided more context to the statement in the manuscript. We were prompted to use the TET inducible system to determine the relationship between protein expression and mitochondrial cluster formation. Most conditions that have been reported to cause mitochondrial perinuclear clusters use transient transfection or retroviral transduction, where the protein would be present for >24 hours before analysis. On this scale, we consider four hours to be rapid. For additional context, we say rapid as we have other proteins that cause perinuclear mitochondrial clusters and it takes 16-24 hours for the change in mitochondrial distribution in the same TET inducible system.

We also have added Western blot analysis of protein expression and quantification of mitochondrial clustering in cells treated with TET for 2, 3, 4, 5 or 6 hours to Figure S4.

It was a bit curious that only the MFN1 variants were examined in Figures 7 & 8, and that nothing was done for the MFN2 variants. Is there any specific reason for this, or is it just that the MFN1 and MFN2 variants behaved primarily the same way in earlier experiments?

We focused our analysis on Mfn1 in the final figures as most of our analysis indicated that Mfn1-S329P and Mfn2-S350P were more similar than different.

I'm not sure that I agree with the title stating that the 'variant reveals that mitochondrial tethering requires GTP hydrolysis', as it implies the variant does not have GTPase activity or tethering abilities. In fact, their data show the opposite, that the variant has GTPase activity, and it leads to excessive tethering. It is only when they introduce additional mutations that impair GTPase activity that the excessive tethering is inhibited. Maybe something along the lines of 'Characterization of the S350P MFN2 variant helps to reveal that mitochondrial tethering requires GTP hydrolysis'.

We have modified the title of the manuscript to reflect this important note.

Reviewer #3 (Comments to the Authors (Required)):

In the present study, the authors very nicely describe a novel dominant negative role of mitofusin mutants in perinuclear clustering of mitochondria, advancing our current knowledge of mitochondrial fusion. Mitochondrial clustering depends on both GTP binding and hydrolysis but is not the result of altered mitochondrial transport events. Cytosolic mitofusins efficiently titrated mitochondrial clustering, consistent with a trans tether trapping of mitochondria that prevents further progression through the fusion cycle. Moreover, these variants -located in a critical hinge point between the N-terminally located GTPase domain and the 1st helix bundle- mimic a mutation present in Charcot-Marie-Tooth patients, thus this manuscript also provides some hints for understanding how mitofusin 2 can cause neurodegeneration. This manuscript is largely based on mitochondrial morphology analysis, inferred from biochemical and structural features of the mini-mitofusins. It would be important to support their findings by mechanistic analysis using full length mitofusins, thus allowing to place the function of the mutated residues in the fusion cycle and to confirm the proposed model.

Main concerns:

The authors show that perinuclear clustering depends on GTP binding and hydrolysis, and on the G-G interface. It would be important to show where in the fusion cycle these mutants are blocked:
A - To support their hypothesis that the proline mutations create hyperactive mitofusin tethering variants, the authors should analyse cis and trans binding, at least for Mfn1, with isolated mitochondria and differently tagged mitofusin versions, using all combinations of the proline mutants and previously described mutants used in Fig. 6C and 6E.

We agree that it would be interesting to clearly demonstrate the stage of fusion that is blocked for each of the variants included in Figures 6. Respectfully, we contend that this work is beyond the scope of this manuscript. Here, we use these amino acid substitutions only to determine the role of the GTPase domain in the dominant negative function of Mfn1S329P or Mfn2S350P, not to dissect the role of each of these GTPase variants in mitochondrial fusion.

The question of how each amino acid substitution alters normal mitochondrial fusion is open and important, but we cannot address it here. Figures 6C and 6E described eight different mitofusin variants in the original submission, and now ten in our revised manuscript. It is clear from our work and other published experiments that none of the variants support mitochondrial fusion, yet some are likely to have interesting molecular defects. The characterization of these defects would likely enhance our understanding of the role of the catalytic cycle in mitofusin-mediated membrane tethering and fusion. However, this would represent at least 12 months of work in our lab, if everything worked perfectly. A significant barrier to rapid progress is the fact that this analysis would require the generation of clonal populations of Mfn1-null cells expressing near-endogenous levels of each variant. In the mixed populations shown in this manuscript, we have many cells that express undetectable levels of protein (see comments above). For microscopy, we can simply not score those cells. However, in mitochondrial preparations, these cells dilute our mitofusin variant significantly in the total population. As such, we have found that we cannot replicate our *trans* co-immunoprecipitation experiment with cells from a mixed population due to low protein abundance (Engelhart & Hoppins 2019 and Samanas et al., 2020).

B - Similarly, in vitro fusion assays with isolated mitochondria should be performed on all mutation combinations as in Fig. 6C and 6E for Mfn1.

This analysis would also require that we generate clonal populations for every mitofusin variant described in Figure 6. We attempted to make clonal populations of Mfn1-null cells expressing Mfn1-S329P years ago and failed after multiple attempts. We believe that the constitutive expression of Mfn1-S329P and the resulting mitochondrial perinuclear clusters impairs mitochondrial segregation during mitosis. We have captured cell division events in a population of cells expressing Mfn1-S329P and observed most mitochondria distributing to one daughter cell, leaving the second cell with no mitochondria. Example images are included below. Therefore, we cannot perform our in vitro mitochondrial fusion assay with the proline variants.

Most cells expressing the mitofusin variants with substitutions in the GTPase domain have fragmented mitochondrial networks, consistent with severely impaired mitochondrial fusion activity. Thus, knowledge gained from this assay is not expected to change the interpretation of this manuscript. We respectfully assert that in vitro fusion for 12 cell lines is also beyond the scope of this manuscript.

C- As suggested by the authors, irregular mitochondrial localization of the mitofusin variants could indicate failure in complex disassembly, like previously shown for yeast mitofusin variants (Anton et. al. 2020). Thus, the authors could quantify mitofusins loci formation, and also analyse perinuclear mitochondrial clustering of their proline mutant, upon simultaneous inhibition of mitochondrial fission, to force mitochondrial tubulation.

We have found quantitative analysis of the mitofusin foci to be challenging. Qualitatively, there are more and larger foci in cells with perinuclear clusters; however, given that the mitochondria cannot be resolved, it is likely that we cannot resolve individual foci. Thus, we chose to remove our comment from the discussion.

To inhibit mitochondrial fission, we expressed Drp1-GFP-G350D in the HEK293 Flp-In TReX cells with Mfn1-Flag-S329P integrated at the TET-controlled promoter. As a control, we expressed Drp1-GFP-WT. As with the experiments described in Figure 8, 3×10^5 cells were first seeded on a microscope dish. Approximately 24 hours later, the cells were transfected with 300 ng Drp1-GFP DNA using JetPrime transfection reagent. The following day, cells were incubated with Mitotracker CMXRos and NucBlue to stain mitochondria and nuclei, respectively. Cells were either incubated with or without 0.2 micrograms/mL TET for four hours and then imaged by fluorescence microscopy. The mitochondrial structure was scored in cells with GFP signal. The results of three independent experiments are shown below:

We observed no change in the percent of cells that have perinuclear mitochondrial clusters when expression of Mfn1-S329P is induced in cells expressing Drp-GFP-WT or G350P. As expected, when expression of Mfn1-S329P is not induced, there was an increase in hyperfused mitochondrial networks when Drp1-GFP-G350D was expressed, but not Drp1-GFP-WT.

D-In Fig 8 the authors show that expression of cytosolic eGFP-Mfn prevents mitochondrial clustering by S329P. However, if the presumption that S3329P prevents Mfn complexes disassembly is correct, one could expect to see co-localization of eGFP-Mfn to mitochondrial-located S329P, especially given its dominant negative properties. Please comment.

It is difficult to predict the behavior of the cytosolic mitofusin given that it lacks the physical restrictions of being embedded in the membrane. The cytosolic protein may be able to make conformational changes that release it from its interaction with membrane bound mitofusin while these changes are not possible when inserted into opposite mitochondrial membranes and engaged in a trans complex.

E-Fig. 7 does not allow concluding that only cis-oligomerization is prevented, given that mini-mfn properties do not prove that the same will occur for the S329P variant once in the full length and membrane contexts. I would suggest to either analyse if the 200 kDa form corresponds to a cis or trans dimer, or instead eliminate this figure.

The BN-PAGE shown in Figure 7 is performed with solubilized mitochondria and thus represents the behavior of full-length mitofusin. We interpret these assemblies as cis because they are observed in the apo- and GMPPNP-bound state. In contrast, mitofusin trans complex assembly has only been reported when bound to the transition state mimic. Furthermore, given that we see most of the ~200 kDa disappear upon incubation with GMPPNP, we favor interpreting these assemblies as mitofusin in the same membrane as mitofusin-mediated tethering is reported to be a rare and transient event for wild-type protein.

Other points:

-This study shows the importance of hinge 2 for mitofusin activity. However, this hinge has been extensively studied mechanistically, mostly in yeast, from studies performed by the Escobar and Cohen groups. Thus, the previous findings should be clearly cited. Moreover, the authors should clarify how this proline might differ or act similarly to previously described functions in this hinge.

We regret this oversight and have modified the discussion.

“The importance Hinge 2 conformational dynamics is conserved as the yeast outer membrane fusion protein, Fzo1, also requires this domain for fusion activity (De Vecchis et al., 2017, Anton et al., 2019). Specifically, amino acid substitutions that disrupt the salt-bridge interactions between the GTPase domain and HB1 in the closed state abolish Fzo1-mediated fusion.”

-The authors nicely show that the mitochondrial clustering observed in the proline mutant depends on GTP binding and hydrolysis and "propose that mitofusin exists primarily in a low-energy state that is inactive and that GTP hydrolysis converts mitofusin to a high-energy state that can tether mitochondria." It has been previously shown that GTP binding is necessary for mitofusin complex formation in cis and in trans, in contrast to GTP hydrolysis. Thus, it would be important to clearly discuss these differences. Further, the exact "energy state", as it has been extensively studied for viral fusion proteins, has not been shown in this study. So, perhaps the authors could clarify what they mean.

Given the result that Mfn1_{IM}-H107A does not form dimers, we have modified our model and most of this language has been removed from the manuscript.

- The authors claim that "the mitochondria in cells expressing the mitofusin proline variant possessed cristae structures similar to wild-type, consistent with normal mitochondrial function". However mitochondrial function is not shown/characterised. Please clarify.

Cristae structure is sensitive to changes in mitochondrial function. The fact that the cristae are comparable in wild type controls and cells expressing Mfn1-S329P suggest that gross mitochondrial dysfunction does not result from expression of this variant. We have clarified this point in the text.

Minor comment:

-"To determine enzymatic activity of Mfn1S329P, we used the internally modified Mfn1 (Mfn1IM) that consists of residues 1-365 and 696-741 connected by a flexible linker (Cao et al., 2017). We found the kinetics of GTP hydrolysis was similar between Mfn1IM WT and Mfn1IM S329P (Fig. 6, A & B)" For easier understanding the authors could clarify here that Mfn1IM corresponds to "mini-mitofusin 1".

We have made this change in the text.

August 30, 2022

RE: Life Science Alliance Manuscript #LSA-2021-01305-TR

Dr. Suzanne Hoppins
University of Washington
Biochemistry
1959 NE Pacific St Health Sciences Building J383
Seattle, WA 98195

Dear Dr. Hoppins,

Thank you for submitting your revised manuscript entitled "A dominant negative mitofusin causes mitochondrial perinuclear clusters due to aberrant tethering". We would be happy to publish your paper in Life Science Alliance pending final revisions necessary to meet our formatting guidelines.

- please address Reviewer 2's remaining comment
- please provide your manuscript as an editable doc file
- please consult our manuscript preparation guidelines <https://www.life-science-alliance.org/manuscript-prep> and make sure your manuscript sections are in the correct order
- please add a category for your manuscript in our system
- please add the Twitter handle of your host institute/organization as well as your own or/and one of the authors in our system
- please add the author contributions and a conflict of interest statement to the main manuscript text
- please use the [10 author names, et al.] format in your references (i.e. limit the author names to the first 10)
- please double-check your figure callouts and figure legend for Figure 5; you have a callout for Figure 5E, but this is not in the legend or in the figure
- please add a callout for Figure 5C, Figure 7F and Figure S3 to your main manuscript text

A. FINAL FILES:

B. MANUSCRIPT ORGANIZATION AND FORMATTING:

Sincerely,

Reviewer #1 (Comments to the Authors (Required)):

The authors have satisfactorily addressed my concerns.

Reviewer #2 (Comments to the Authors (Required)):

Overall, I'm satisfied with the revised manuscript and response to the reviews. The only minor comment I have is that the labels for the wild-type and -/- Mefs in the PA-GFP assays seems to be mixed-up (figs 3C, 3D, and S9). I think the grey bars should be the WT Mefs and the black bars should be the -/- Mefs.

Reviewer #3 (Comments to the Authors (Required)):

The authors have sufficiently addressed the reviewing suggestions.

We contend that the outlined time and effort required in the generation of clonal populations of Mfn1 mutant cells, necessary for a deeper understanding of the described mitofusin mutant variants, go beyond the constraints of this study. Similarly, we appreciate the technical difficulties in the analysis of mitofusin foci.

September 28, 2022

RE: Life Science Alliance Manuscript #LSA-2021-01305-TRR

Dr. Suzanne Hoppins
University of Washington
Biochemistry
1959 NE Pacific St Health Sciences Building J383
Seattle, WA 98195

Dear Dr. Hoppins,

Thank you for submitting your Research Article entitled "A dominant negative mitofusin causes mitochondrial perinuclear clusters due to aberrant tethering". It is a pleasure to let you know that your manuscript is now accepted for publication in Life Science Alliance. Congratulations on this interesting work.

DISTRIBUTION OF MATERIALS:

Again, congratulations on a very nice paper. I hope you found the review process to be constructive and are pleased with how the manuscript was handled editorially. We look forward to future exciting submissions from your lab.

Sincerely,
